# A tryparedoxin-coupled biosensor reveals a mitochondrial trypanothione metabolism in trypanosomes

**Samantha Ebersoll[1], Marta Bogacz[1], Lina M Günter[1], Tobias P Dick[2], R Luise Krauth-Siegel[1]***

[1]Biochemie-Zentrum der Universität Heidelberg, Heidelberg, Germany; [2]Division of Redox Regulation, DKFZ-ZMBH Alliance, German Cancer Research Center (DKFZ), Heidelberg, Germany

**Abstract** Trypanosomes have a trypanothione redox metabolism that provides the reducing equivalents for numerous essential processes, most being mediated by tryparedoxin (Tpx). While the biosynthesis and reduction of trypanothione are cytosolic, the molecular basis of the thiol redox homeostasis in the single mitochondrion of these parasites has remained largely unknown. Here we expressed Tpx-roGFP2, roGFP2-hGrx1 or roGFP2 in either the cytosol or mitochondrion of *Trypanosoma brucei*. We show that the novel Tpx-roGFP2 is a superior probe for the trypanothione redox couple and that the mitochondrial matrix harbors a trypanothione system. Inhibition of trypanothione biosynthesis by the anti-trypanosomal drug Eflornithine impairs the ability of the cytosol and mitochondrion to cope with exogenous oxidative stresses, indicating a direct link between both thiol systems. Tpx depletion abolishes the cytosolic, but only partially affects the mitochondrial sensor response to $H_2O_2$. This strongly suggests that the mitochondrion harbors some Tpx and, another, as yet unidentified, oxidoreductase.

**\*For correspondence:**
luise.krauth-siegel@bzh.uni-heidelberg.de

**Competing interests:** The authors declare that no competing interests exist.

## Introduction

Trypanosomatids are the causative agents of African sleeping sickness (*Trypanosoma brucei gambiense* and *T. b. rhodesiense*), Nagana cattle disease (*Trypanosoma b. brucei*, *T. congolense*), South-American Chagas' disease (*T. cruzi*) and the different manifestations of leishmaniasis (*Leishmania* spp.). All these parasitic protozoa lack the nearly ubiquitous glutathione reductases (GRs) and thioredoxin reductases but, instead, have a trypanothione (T(SH)$_2$)/trypanothione reductase (TR) system. The trypanothione system delivers reducing equivalents for a large number of vital processes (for reviews see *Krauth-Siegel and Leroux, 2012*; *Manta et al., 2018* and *Manta et al., 2013*). Most of the reactions are mediated by tryparedoxin (Tpx), a distant member of the thioredoxin family.

For T(SH)$_2$ synthesis, two molecules of glutathione (GSH) are linked by one molecule of spermidine, with glutathionylspermidine (Gsp) as intermediate. Both steps are catalyzed by trypanothione synthetase (TryS) (*Comini et al., 2004*; *Krauth-Siegel and Leroux, 2012*; *Oza et al., 2002*). Difluoromethylornithine (DFMO, Eflornithine), a drug against late-stage *T. b. gambiense* sleeping sickness, is an irreversible inhibitor of ornithine decarboxylase (ODC), the enzyme generating putrescine, the precursor for spermidine synthesis. Treatment of *T. brucei* with DFMO decreases the levels of spermidine and T(SH)$_2$ and slightly increases the level of GSH (*Bellofatto et al., 1987*; *Fairlamb et al., 1987*; *Xiao et al., 2009*). All enzymes involved in T(SH)$_2$ biosynthesis and also TR, which catalyses the NADPH-dependent reduction of trypanothione disulfide (TS$_2$) as well as glutathionylspermidine disulfide (Gsp$_2$), appear to be restricted to the cytosol.

African trypanosomes lack catalase. Hydroperoxides are detoxified by 2-Cys-peroxiredoxins (Prxs) (*Budde et al., 2003*; *Tetaud et al., 2001*; *Wilkinson et al., 2003*) and non-selenium glutathione

**eLife digest** *Trypanosoma brucei* are single-celled parasites that cause human sleeping sickness and animal diseases. Like in other organisms, the parasite contains different compartments, each having several specific roles. The mitochondrion is the compartment that provides most of the energy needed to keep the cell alive.

Many cellular processes, such as those that happen in the mitochondrion, produce compounds including hydrogen peroxide that can cause 'oxidative damage'. To counteract this, cells make small molecules called thiols. These thiols provide 'reducing' power to chemically balance out the oxidative damage. Trypanosomes have an unusual thiol system that relies on a molecule called trypanothione. *Trypanosoma brucei* cells make trypanothione in the cytosol, the fluid which surrounds all cellular compartments; here it is also used up with the help of a protein called tryparedoxin. However, it was not known which thiols are present in the mitochondrion.

Ebersoll et al. have now made a molecular sensor that can detect trypanothione. The sensor includes a fluorescent protein, which changes its brightness based on its oxidation state, fused to the tryparedoxin protein. This probe could either be put in the cytosol or mitochondrion of *Trypanosoma brucei* cells. Treating the cells with hydrogen peroxide changed the fluorescence of the biosensor. *Trypanosoma brucei* cells without tryparedoxin protein in their cytosol still responded to an oxidative challenge in the mitochondrion. The experiments reveal that trypanosomes do have a mitochondrial trypanothione system.

This new fluorescent biosensor will be used to study how other cellular compartments deal with oxidative conditions. The tests will reveal how different compartments communicate with each other to counteract the stress. The sensor could also be used to determine how anti-parasite drugs affect the cells' trypanothione system.

peroxidase-type enzymes (Pxs) (*Hillebrand et al., 2003*; *Schlecker et al., 2005*; *Wilkinson et al., 2003*). The Pxs preferably reduce lipid hydroperoxides (*Bogacz and Krauth-Siegel, 2018*; *Diechtierow and Krauth-Siegel, 2011*; *Hiller et al., 2014*; *Schaffroth et al., 2016*) whereas the Prxs mainly detoxify hydrogen peroxide and peroxynitrite (*Thomson et al., 2003*; *Trujillo et al., 2004*). Both types of thiol peroxidases are reduced by the TR/T(SH)$_2$/Tpx system and thus act as tryparedoxin peroxidases (*Scheme 1*; *Castro and Tomás, 2008*; *Krauth-Siegel and Comini, 2008*; *Krauth-Siegel and Leroux, 2012*; *Manta et al., 2013*).

*T. brucei* encodes three virtually identical Pxs (*Hillebrand et al., 2003*) which occur in the cytosol (Px I and II) and the mitochondrion (Px III). Depletion or deletion of the Pxs is lethal (*Diechtierow and Krauth-Siegel, 2011*; *Hiller et al., 2014*; *Schaffroth et al., 2016*; *Schlecker et al., 2005*; *Wilkinson et al., 2003*). However, survival and proliferation of cells lacking Pxs can be restored by supplementing the medium with an iron chelator or the vitamin E analogue Trolox [(±)−6-hydroxy-2,5,7,8-tetramethylchromane-2-carboxylic acid] (*Bogacz and Krauth-Siegel, 2018*; *Diechtierow and Krauth-Siegel, 2011*; *Hiller et al., 2014*; *Schaffroth et al., 2016*). Cell death of Px-deficient insect stage procyclic (PC) *T. brucei* closely resembles ferroptosis in mammalian cells (*Bogacz and Krauth-Siegel, 2018*). In the case of the Prxs, two different enzymes are expressed, one in the cytosol (cPrx) and one in the mitochondrion (mPrx) (*Tomás and Castro, 2013*). In bloodstream (BS) *T. brucei*, depletion of cPrx, but not of mPrx, is lethal (*Wilkinson et al.,*

**Scheme 1.** Reduction of hydroperoxides (ROOH) to the respective alcohol (ROH) by Px- and Prx-type enzymes in *T. brucei*. The reduced form of the thiol peroxidases is restored by interaction with the Tpx/T(SH)$_2$ couple followed by TR and NADPH as final electron source.

*2003*). In *Leishmania*, mPrx is required for survival of the parasite in the mammalian host. However, this essential function is not related to its peroxidase activity but to the ability of the protein to act as a thiol-independent molecular chaperone (*Castro et al., 2011*; *Teixeira et al., 2015*).

The thiol-disulfide redox state of roGFP2 (reduction-oxidation-sensitive green fluorescence protein 2) can be monitored in real time, inside intact cells, in a ratiometric manner (*Meyer et al., 2007*; *Meyer and Dick, 2010*; *Schwarzländer et al., 2016*). Importantly, this approach allows to measure the degree of probe oxidation independently of absolute sensor concentration. Equilibration of the roGFP2 thiol-disulfide redox couple with the intracellular glutathione redox couple was found to be mediated, almost exclusively, by glutaredoxins (Grxs) (*Meyer et al., 2007*). Thus, fusion of roGFP2 to human glutaredoxin 1 (hGrx1) generated a probe for specifically measuring the redox potential of the glutathione redox couple ($E_{GSH}$) which is a function of both the GSH:GSSG ratio and total glutathione concentration (*Gutscher et al., 2008*; *Morgan et al., 2013*; *Morgan et al., 2011*). The hGrx1-roGFP2 fusion protein proved to be a valuable tool for measuring changes in $E_{GSH}$ in various model organisms and compartments (*Gutscher et al., 2008*; *Kojer et al., 2012*; *Morgan et al., 2013*; *Radzinski et al., 2018*). It has also been used to follow drug-induced changes of $E_{GSH}$ in the malaria parasite *Plasmodium falciparum* (*Kasozi et al., 2013*) and in *T. brucei* (*Franco et al., 2017b*; *Franco et al., 2017a*). By genetically fusing roGFP2 to thiol-disulfide oxidoreductases other than Grx, additional biosensors, selective for other kinds of small thiols, were generated. For instance, fusion of roGFP2 to mycoredoxin-1 yielded a probe selectively responding to changes in the mycothiol redox potential in actinomycetes (*Bhaskar et al., 2014*; *Reyes et al., 2018*; *Tung et al., 2019*).

In yeast and human cells, GSH biosynthesis is confined to the cytosol. GSH must therefore be imported into both the intermembrane space (IMS) and the matrix of the mitochondrion (*Calabrese et al., 2017*). In yeast, the hGrx1-roGFP2 probe revealed that the cytosol, IMS and mitochondrial matrix rapidly recover after exposure of cells to an oxidant (*Kojer et al., 2015*). This indicates that all three compartments either harbor a GSSG reducing system or are connected to a compartment in which GSSG is reduced. Indeed, GR localizes to both the cytosol and the mitochondrial matrix (*Calabrese et al., 2017*). Trypanosomatids contain both free GSH and T(SH)$_2$ (*Ariyanayagam and Fairlamb, 2001*; *Krauth-Siegel and Comini, 2008*). This finding, however, does not provide any information about the nature and concentration of the individual thiol(s) within distinct subcellular compartments. Trypanosomes have a single mitochondrion that spans the complete length of the cell. In BS *T. brucei*, the mitochondrion has a tubular cristae-poor morphology and the parasites produce ATP by glycolysis and possibly also by substrate-level phosphorylation reactions in the mitochondrial matrix (*Zíková et al., 2017*). In contrast, PC cells have a highly reticulated, cristae-rich mitochondrion and generate ATP via oxidative phosphorylation. PC *T. brucei* have a high demand for mitochondrial iron sulfur cluster proteins (*Lukeš and Basu, 2015*), the synthesis of which usually involves GSH (*Lill et al., 2015*). However, virtually nothing is known about the low-molecular-weight thiols that reside in the mitochondrion of trypanosomatids (*Manta et al., 2013*; *Tomás and Castro, 2013*).

Here we studied PC *T. brucei* cell lines constitutively expressing a novel biosensor, Tpx-roGFP2, in either the cytosol or mitochondrion. We show for the first time that the mitochondrial matrix of trypanosomes harbors a trypanothione-based redox metabolism. The disulfide reducing capacity in the mitochondrial matrix appears to be slightly weaker than in the cytosol, probably due to a lower T(SH)$_2$/TS$_2$ ratio. Inhibition of T(SH)$_2$ biosynthesis by DFMO does not alter the redox steady state of the probe but impairs the ability of both the cytosol and mitochondrion to cope with exogenously applied oxidants, indicating a direct link between the trypanothione systems of the two compartments. Depletion of Tpx abolishes the response of cytosolic roGFP2-hGrx1 or roGFP2 to H$_2$O$_2$ but only partially affects the mitochondrial sensor response. This finding strongly suggests that the mitochondrion harbors low levels of Tpx and, in addition, another, as yet unidentified, oxidoreductase that is also able to transfer reducing equivalents from T(SH)$_2$ to the mitochondrial thiol peroxidases. Notably, our data also reveal that the high selectivity of roGFP2-hGrx1 for the GSH/GSSG redox couple is restricted to cells that do not convert GSH into additional closely related low molecular weight thiols such as Gsp and T(SH)$_2$. In trypanosomes, roGFP2-hGrx1 predominantly equilibrates with the T(SH)$_2$/TS$_2$ redox couple.

## Results

### Tpx-roGFP2 is a sensor for the trypanothione redox state

To generate a trypanothione-selective sensor, roGFP2 was fused to Tpx. Recombinant Tpx-roGFP2, hGrx1-roGFP2 and roGFP2 were purified by metal affinity chromatography (*Figure 1—figure supplement 1*). The pre-reduced proteins were treated with GSSG, $Gsp_2$ or $TS_2$ and the degree of oxidation (OxD) of the probe was followed over time (*Figure 1—figure supplement 2*). Representative unprocessed fluorescence curves are shown in *Figure 1—figure supplement 3*. At an equimolar concentration of 1 µM, Tpx-roGFP2 responded slightly faster to $Gsp_2$ and $TS_2$ than to GSSG (*Figure 1A–C*). As previously published, hGrx1-roGFP2 was rapidly oxidized by GSSG (*Gutscher et al., 2008*; *Figure 1D*). The same sensor was also efficiently oxidized by $Gsp_2$ (*Figure 1E*), whereas a comparable reaction with $TS_2$ required a 10-times higher concentration (*Figure 1F* and *Figure 1—figure supplement 2F*), possibly due to the steric constrains imposed by the cyclic disulfide. Unfused roGFP2 did not react efficiently with any of the disulfides (at 1 µM) and at high concentrations was more efficiently oxidized by $Gsp_2$ and $TS_2$ than by GSSG (*Figure 1G–I* and *Figure 1—figure supplement 2G–I*). In conclusion, the fusion to either Tpx or hGrx1 accelerated roGFP2 oxidation by all three disulfides.

In the next step, we studied the reactivity of the oxidized probes towards GSH, Gsp, and $T(SH)_2$ (*Figure 2* and *Figure 2—figure supplement 1*). Representative unprocessed fluorescence curves are shown in *Figure 2—figure supplement 2*. Sensor proteins stored in the absence of reducing agent were fully oxidized and thus used without further treatment. Tpx-roGFP2 was readily reduced by 100 µM Gsp and 10 µM $T(SH)_2$ whereas 100 µM GSH did not cause any reaction (*Figure 2A–C*). Reduction of Tpx-roGFP2 by even 1 mM GSH was very slow. In contrast, 500 µM $T(SH)_2$ resulted in almost complete probe reduction within five min (*Figure 2—figure supplement 1A and C*). In comparison, hGrx1-roGFP2 showed virtually the same low reduction rate when treated with 100 µM GSH, 100 µM Gsp or 10 µM $T(SH)_2$ (*Figure 2D–F*). Clearly, for significant reduction, ≥500 µM of either GSH or $T(SH)_2$ were required (*Figure 2—figure supplement 1D and F*). This finding indicated that the selectivity of hGrx1-roGFP2 for GSH applies to cells with an exclusive glutathione system, but not to Kinetoplastida. Unfused roGFP2 was not even reduced by 15 mM GSH, but fully reduced by 7.5 mM $T(SH)_2$ (*Figure 2—figure supplement 1G and I*). Taken together, all three sensors were most efficiently reduced by $T(SH)_2$.

The cellular concentrations of GSH and $T(SH)_2$ in PC *T. brucei* are approximately 100 µM and 500 µM, respectively (see Figure 4B). Reduction of Tpx-roGFP2 and hGrx1-roGFP2 by 1 µM and 10 µM $T(SH)_2$ was accelerated by the presence of 100 µM or 500 µM GSH (*Figure 2—figure supplement 3A–D*, compare with *Figure 2—figure supplement 1C and F*). However, even 500 µM GSH had a negligible additional effect in the presence of ≥100 µM $T(SH)_2$. Unfused roGFP2 was only slightly reduced by 500 µM $T(SH)_2$ and the simultaneous presence of GSH had no additional effect (*Figure 2—figure supplement 3E and F*). Finally, we followed the response of Tpx-roGFP2 and hGrx1-roGFP2 to small incremental additions of GSSG or $TS_2$ when both GSH and $T(SH)_2$ were present at physiological concentrations. In the presence of 100 µM GSH and 500 µM $T(SH)_2$, Tpx-roGFP2 and hGrx1-roGFP2 reached an OxD of 0.15 and 0.24, respectively (*Figure 2—figure supplement 4*). This was in agreement with the data obtained for the individual thiols (*Figure 2—figure supplement 1A, C,D and F*) and supported the concept that $T(SH)_2$ is the superior disulfide reductant for both kinds of sensor. Addition of 1 µM GSSG or $TS_2$ did not have any effect. At a starting OxD of about 0.2, the incremental addition of 1 µM disulfide may be too small to cause any significant further sensor oxidation. When 25 µM disulfide was applied, $TS_2$ oxidized both sensors to a higher degree than GSSG. Interestingly, when treated with 10 µM GSSG, hGrx1-roGFP2 displayed a small transient increase in OxD. Thus, hGrx1-roGFP2 appears to be able to sense GSSG before it rapidly equilibrates with the trypanothione redox couple. Nevertheless, a physiological situation that could give rise to an increase of GSSG, but not of $TS_2$, appears to be highly unlikely. Taken together, under in vitro conditions, Tpx-roGFP2 and hGrx1-roGFP2 displayed a very similar behavior and primarily responded to changes in the $T(SH)_2/TS_2$ ratio.

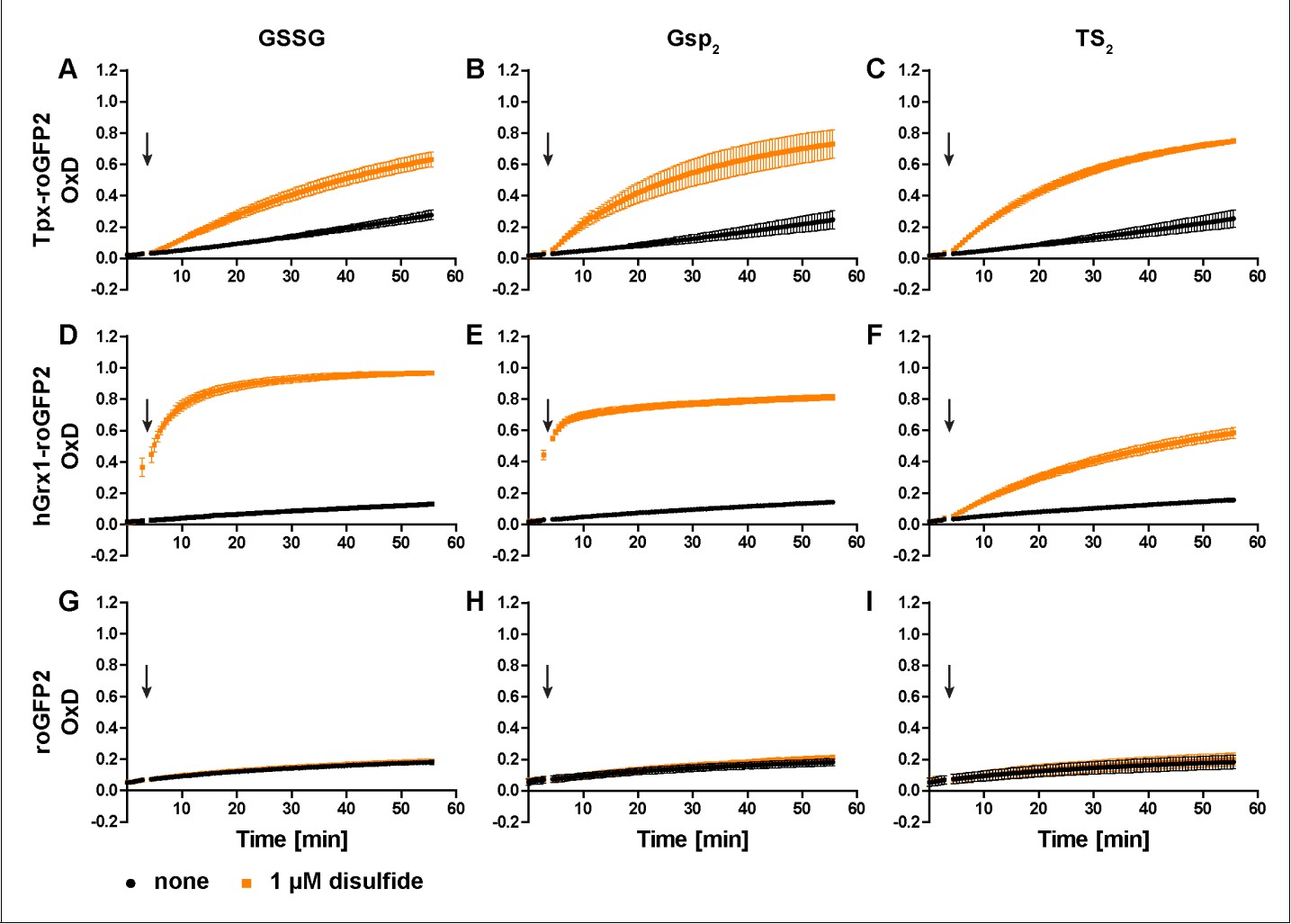

**Figure 1.** Oxidation of the recombinant roGFP2-based sensors by an equimolar concentration of GSSG, $Gsp_2$, or $TS_2$. 1 µM of pre-reduced recombinant (A-C) Tpx-roGFP2, (D–F) hGrx1-roGFP2, and (G–I) roGFP2 was subjected to plate reader-based fluorescence measurements. After 3 min, 1 µM (A, D, G) GSSG, (B, E, H) $Gsp_2$, or (C, F, I) $TS_2$ or the same volume of buffer (none) was added (arrow). The data represent the mean ± standard deviation (SD) of three independent experiments.

The online version of this article includes the following source data and figure supplement(s) for figure 1:

**Figure supplement 1.** Purification of the recombinant sensors.

**Figure supplement 2.** Oxidation of recombinant roGFP2-sensors by different concentrations of GSSG, $Gsp_2$, and $TS_2$.

**Figure supplement 3.** Unprocessed fluorescence intensity curves for 1 µM pre-reduced (A) Tpx-roGFP2, (B) hGrx1-roGFP2 and (C) roGFP2 treated with 1 µM $TS_2$ (arrow).

**Figure supplement 3—source data 1.** Original data for *Figure 1—figure supplement 3*.

## In vivo, the cytosolic sensors are fully reduced and respond to exogenously applied oxidants

To follow probe redox changes in vivo, we generated cell lines constitutively expressing either Tpx-roGFP2, roGFP2-hGrx1 or roGFP2 in the cytosol of the parasites (*Figure 3—figure supplement 1*). The domain order of the hGrx1 fusion protein was inverted, to ensure proper folding (*Figure 3—figure supplement 1C*, see Materials and methods for details). Live cell imaging displayed an intense fluorescence that was equally distributed over the whole cell body in accordance with a cytosolic localization (*Figure 3A*). Under normal culture conditions, all three sensors showed an OxD of close to zero indicating that they are almost fully reduced in the cytosolic environment (*Figure 3B–G*).

Exposure of the parasites to exogenous $H_2O_2$ resulted in a concentration-dependent oxidation of the probes. When 5 µM $H_2O_2$ was applied, Tpx-roGFP2 was oxidized within seconds to an OxD of

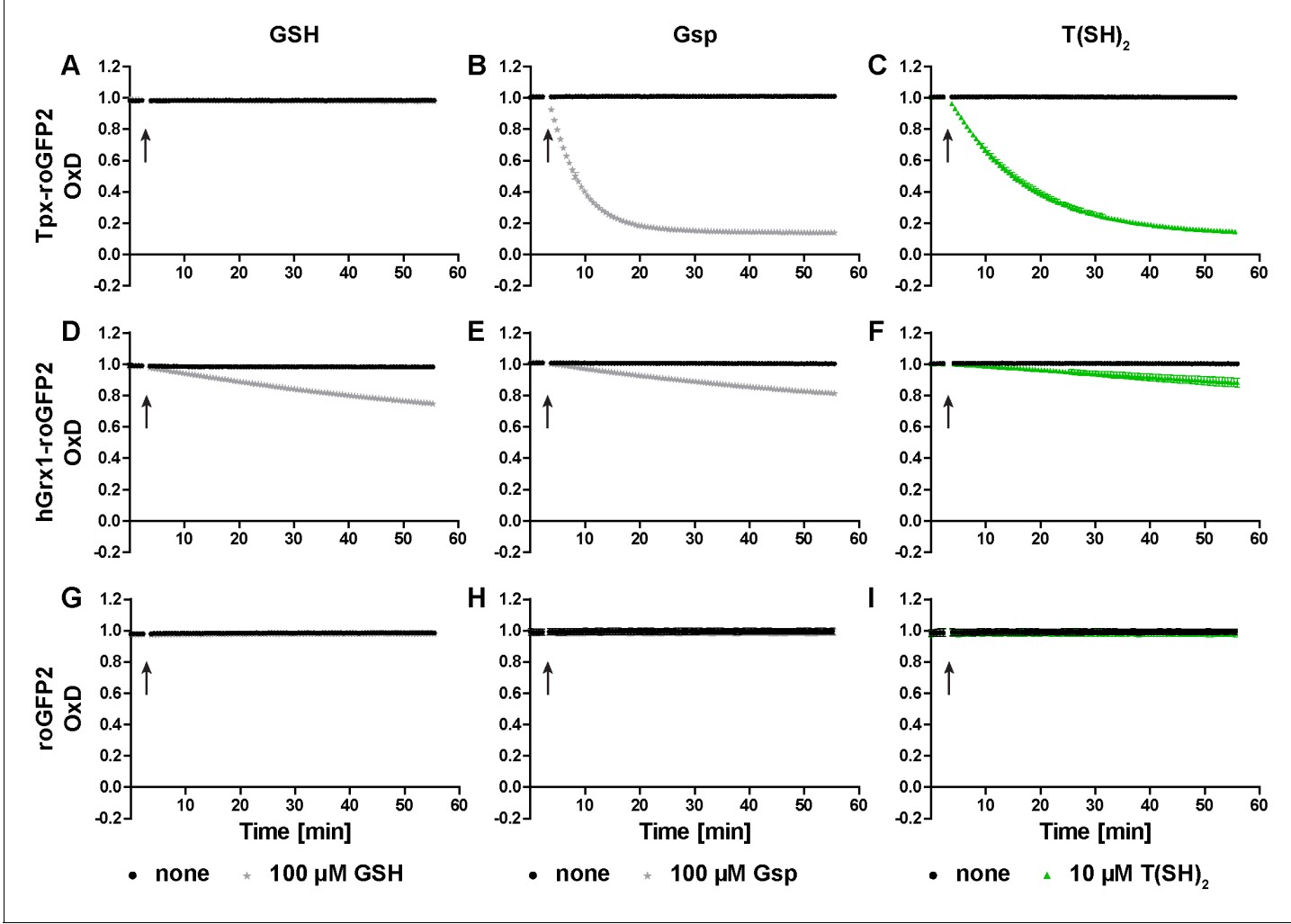

**Figure 2.** Reduction of the roGFP2-based sensors by GSH, Gsp, and T(SH)$_2$. (A–C) Tpx-roGFP2, (D–F) hGrx1-roGFP2, and (G–I) roGFP2 were subjected to fluorescence measurements. After 3 min, (A, D, G) 100 µM GSH, (B, E, H) 100 µM Gsp, (C, F, I) 10 µM T(SH)$_2$, or the same volume of buffer (none) was added (arrow). To keep the thiols reduced during the experiment, the reaction mixtures were supplemented with (A, D, G) 5 mM NADPH and 300 mU hGR or (B, C, E, F, H, I) 0.5 mM NADPH and 30 mU TR. Data are the mean ± SD of three independent experiments except for the ones with 100 µM GSH, which are the mean of two measurements.

The online version of this article includes the following source data and figure supplement(s) for figure 2:

**Figure supplement 1.** Reduction of roGFP2 sensors by different concentrations of GSH, Gsp, or T(SH)$_2$.

**Figure supplement 2.** Unprocessed fluorescence intensity curves for 1 µM (A) Tpx-roGFP2, (B) hGrx1-roGFP2 and (C) roGFP2 treated with 10 µM T(SH)$_2$ (arrow).

**Figure supplement 2—source data 1.** Original data for *Figure 2—figure supplement 2*.

**Figure supplement 3.** T(SH)$_2$ and GSH act additively upon sensor reduction.

**Figure supplement 4.** In the presence of GSH and T(SH)$_2$, Tpx-roGFP2 and hGrx1-roGFP2 are preferably oxidized by TS$_2$ compared to GSSG.

0.8 (*Figure 3B*). Oxidation of roGFP2-hGrx1 and roGFP2 was slightly slower and yielded a maximum OxD of about 0.6 and 0.4, respectively (*Figure 3D and F*). Also, the re-reduction was fastest in the case of Tpx-roGFP2. Nine min after treating the cells with 10 µM H$_2$O$_2$, the sensor was again fully reduced, whereas roGFP2-hGrx1 and roGFP2 required 15 min and 22 min, respectively. The unprocessed fluorescence curves are provided in *Figure 3—figure supplement 2*. Both Tpx-roGFP2 and roGFP2-hGrx1 showed a stronger response than roGFP2, confirming that the detection of transient redox changes by the probes is facilitated by the fused oxidoreductase. In contrast, the response of roGFP2 requires the interaction with an endogenous oxidoreductase, most probably Tpx. Making the assumption that the thiol concentration in the cytosol is comparable to that in the total cell lysate

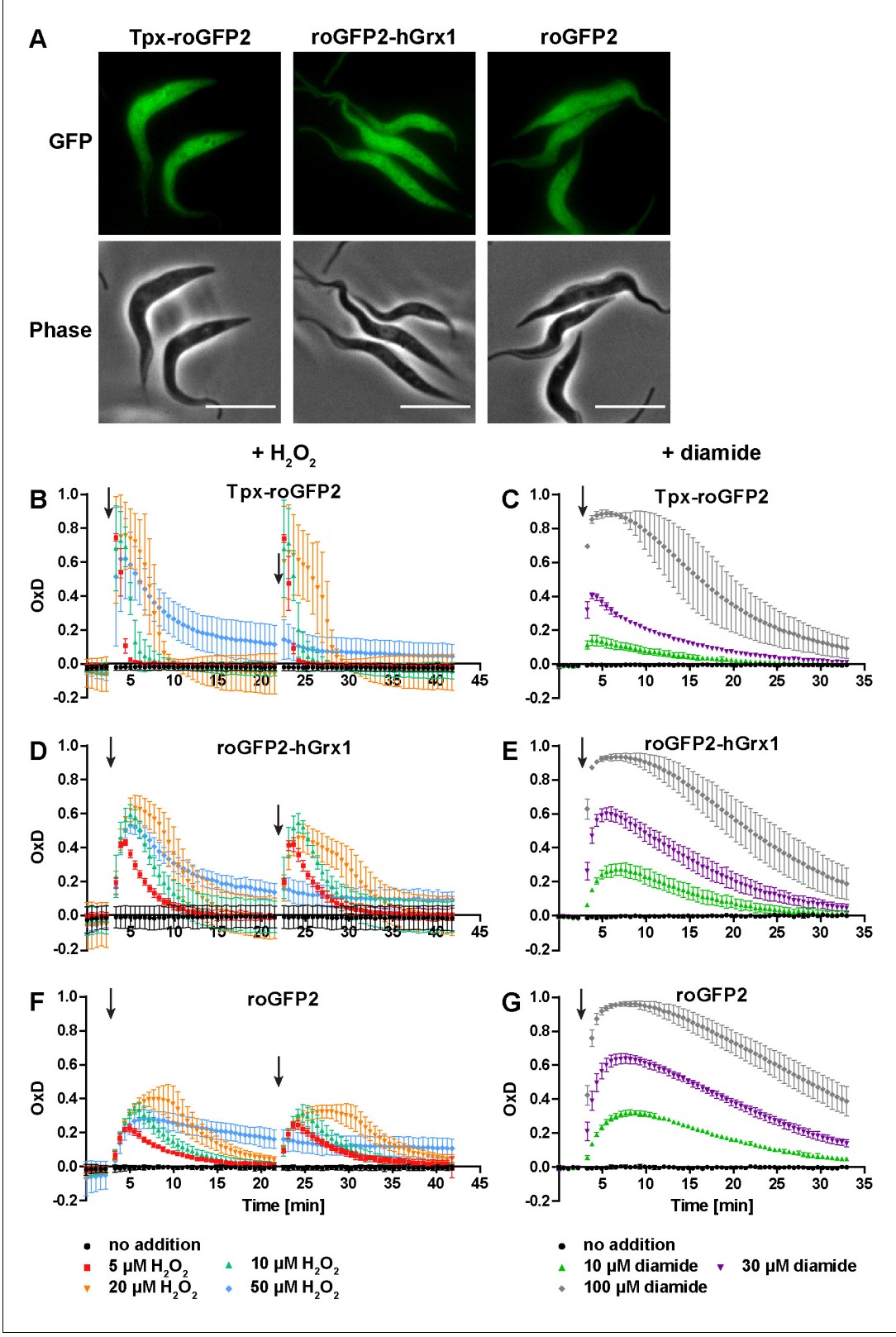

**Figure 3.** The cytosolic sensors are fully reduced and get reversibly oxidized upon treating the parasites with $H_2O_2$ or diamide. (**A**) Live cell imaging of Tpx-roGFP2-, roGFP2-hGrx1-, and roGFP2-expressing PC *T. brucei*. Phase: phase contrast image. Scale bar: 10 μm. (**B–G**) $1 \times 10^7$ PC *T. brucei* constitutively expressing (**B, C**) Tpx-roGFP2, (**D, E**) roGFP2-hGrx1, or (**F, G**) roGFP2 in the cytosol were analyzed in the plate reader based assay. (**B, D, F**) After

*Figure 3 continued on next page*

*Figure 3 continued*

3 min and 22 min, the indicated concentration of $H_2O_2$ was injected (arrow). (C, E, G) Diamide was added after 3 min (arrow). The data are the mean ± SD of at least three individual experiments.

The online version of this article includes the following source data and figure supplement(s) for figure 3:

**Figure supplement 1.** Generation of PC *T. brucei* cell lines that express tag-free roGFP2 sensors in the cytosol.

**Figure supplement 2.** Unprocessed fluorescence intensity curves for cells expressing (A) Tpx-roGFP2, (B) roGFP2-hGrx1 and (C) roGFP2 that were exposed twice to 10 µM $H_2O_2$ (arrows).

**Figure supplement 2—source data 1.** Original data for *Figure 3—figure supplement 2*.

**Figure supplement 3.** Treatment of recombinant roGFP2 sensors with hydrogen peroxide followed by $TS_2$.

**Figure supplement 4.** In PC *T. brucei* treated with 50 µM $H_2O_2$, the sensors still respond to a diamide stress.

(see *Figure 4B*), we conclude that both fusion probes respond primarily to changes in the $T(SH)_2/TS_2$ redox couple. This is supported by the efficient reduction of Tpx-roGFP2 by low micromolar concentrations of $T(SH)_2$ in vitro (*Figure 2C*) and the inability of 100 µM GSH to reduce Tpx-roGFP2 as well as hGrx1-roGFP2 (*Figure 2A and D*). To rule out direct oxidation of the sensors by $H_2O_2$, we incubated the pre-reduced recombinant sensors with 50 µM or even 100 µM $H_2O_2$ for 11 min which resulted in an only marginal increase in OxD comparable with the non-treated controls (*Figure 3—figure supplement 3*). Upon subsequent addition of $TS_2$, all three sensors were oxidized with the same rate and to the same degree as the respective non-treated probe (*Figure 3—figure supplement 3* compare with *Figure 1—figure supplement 2C, F andI*). The data demonstrated that also the Tpx-roGFP2 sensor is insensitive towards micromolar $H_2O_2$ concentrations and confirmed previous studies on hGrx1-roGFP2 and roGFP2 (*Gutscher et al., 2008*; *Kasozi et al., 2013*; *Müller et al., 2017*).

All three sensors completely recovered when cells were treated with up to 20 µM $H_2O_2$. A second $H_2O_2$ bolus elicited an almost identical response, demonstrating full reversibility of the response. In contrast, after exposing the cells to 50 µM $H_2O_2$, the sensors remained partially oxidized, and a second $H_2O_2$ bolus did not trigger any further oxidation (*Figure 3B, D and F*). However, when diamide was used as a second bolus treatment, all sensors were again oxidized, ruling out irreversible probe inactivation by hyperoxidation (*Figure 3—figure supplement 4*). Most likely, the cPrx was hyperoxidized and inactivated by the high $H_2O_2$ concentration, thus abrogating the Prx-catalyzed generation of $TS_2$ responsible for oxidizing the roGFP2-based probes.

When the cells were challenged with a single bolus of 10 µM or 30 µM diamide, Tpx-roGFP2 was rapidly oxidized, but to a lower maximum OxD than cells expressing roGFP2-hGrx1 or roGFP2 (*Figure 3C,E G*). All sensors became fully oxidized upon exposure of the cells to 100 µM diamide, suggesting that they were directly oxidized. In general, probe recovery (from comparable OxD maxima) was slower in diamide- than in $H_2O_2$-treated cells. As observed in BS *T. brucei* (*Ulrich et al., 2017*), diamide treatment of the PC cells may primarily lead to the formation of protein mixed disulfides which, most likely, are more slowly reduced than $TS_2$. After both $H_2O_2$- and diamide-stress, re-reduction of Tpx-roGFP2 was fastest, followed by roGFP2-hGrx1 and roGFP2, corroborating the kinetic superiority of the Tpx-coupled sensor also under in vivo conditions.

## Inhibition of trypanothione biosynthesis affects the oxidative stress response of the parasite

Treatment of *T. brucei* with DFMO results in impaired de novo synthesis of $T(SH)_2$ (*Bellofatto et al., 1987*; *Fairlamb et al., 1987*). Wildtype (WT) parasites and Tpx-roGFP2-expressing cells were cultured in the presence or absence of 5 mM DFMO. After 48 hr, DFMO-treated cells displayed a significant proliferation defect (*Figure 4A*) and growth almost stopped after 72 hr. No difference was observed between WT parasites and cells expressing Tpx-roGFP2, ruling out any effect of the sensor on the parasites' sensitivity towards DFMO.

Cellular free thiol concentrations were determined by preparing total lysates, labeling the thiols with monobromobimane (mBBr), and quantifying the fluorescent derivatives by HPLC analysis (*Figure 4B*). When untreated WT cells were lysed in the presence of TCEP, we obtained total GSH and $T(SH)_2$ concentrations of 134 ± 8 µM and 521 ± 34 µM, respectively. In cells cultured for 48 hr in

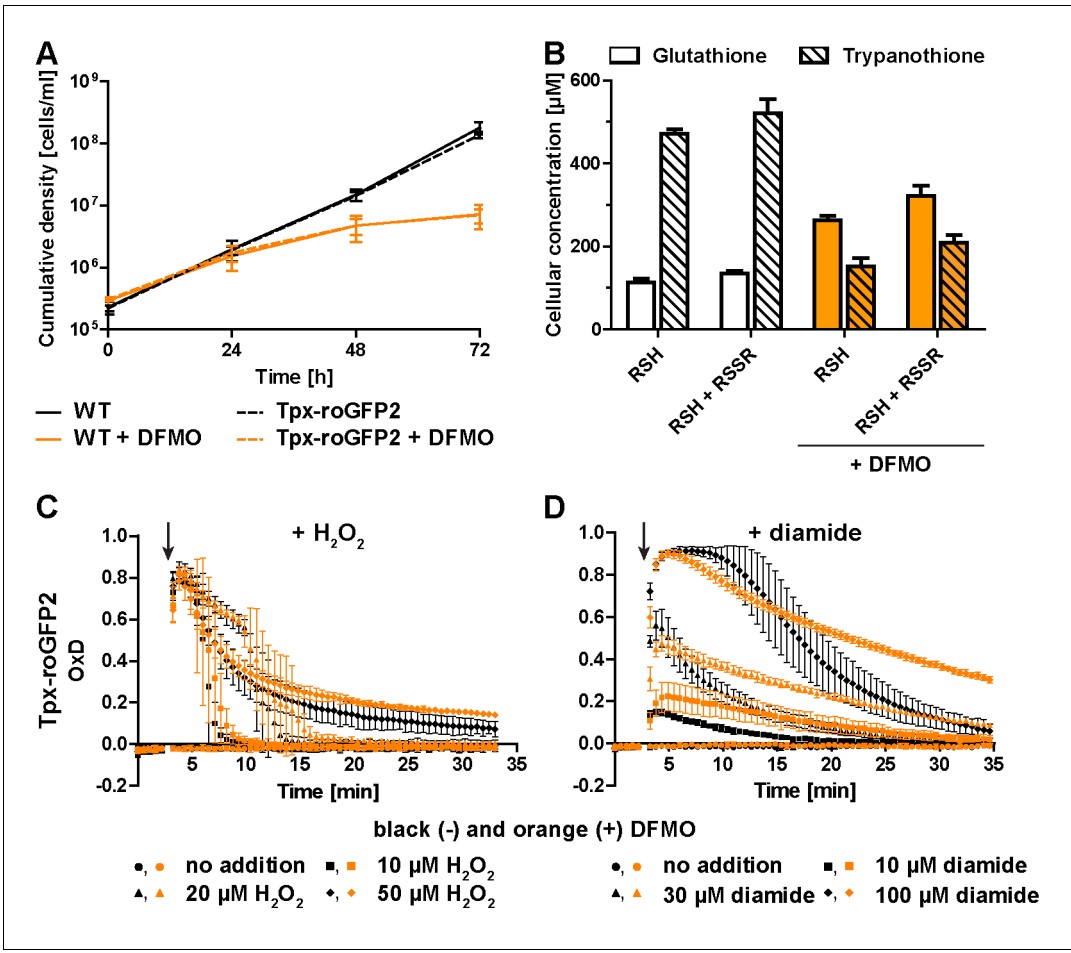

**Figure 4.** Inhibition of T(SH)$_2$ synthesis slows down re-reduction of the oxidized sensor. (**A**) Proliferation of PC WT and Tpx-roGFP2-expressing cells in the absence (black) or presence (orange) of 5 mM DFMO. (**B**) WT parasites grown for 48 hr in medium ±DFMO were lysed by TCA precipitation. The concentration of free thiols was determined either directly (RSH) or after treatment with TCEP yielding the sum of free thiols and disulfides (RSH + RSSR). The values are the mean ± SD of at least three independent analyses. (**C and D**) Tpx-roGFP2-expressing cells that were grown ±DFMO for 48 hr were subjected to fluorescence measurements. After 3 min (arrow), the indicated concentration of (**C**) H$_2$O$_2$ or (**D**) diamide was injected and the sensor response followed over time. Data show the mean ± SD of three independent experiments.

The online version of this article includes the following figure supplement(s) for figure 4:

**Figure supplement 1.** Prolonged cultivation with DFMO for 72 hr does not augment the sensitivity of PC cells towards oxidative stresses.

---

the presence of DFMO, the GSH and T(SH)$_2$ concentrations were 322 ± 24 μM and 210 ± 17 μM, respectively (*Figure 4B*). Thus, in DFMO treated cells, the GSH concentration was doubled, whereas the T(SH)$_2$ concentration was halved, in agreement with published data (*Bellofatto et al., 1987*). When the parasites were disintegrated in the absence of the reducing agent TCEP, the values in all samples were 10–20% lower, indicating that the cellular thiols are mainly in the reduced state as previously shown for BS *T. brucei* (*Ulrich et al., 2017*). This finding matches the highly reduced state of the sensors expressed in the cytosol. Clearly, DFMO treatment did not cause any shift of intracellular thiol/disulfide ratios. The basal OxD of the Tpx-roGFP2 sensor was almost zero, independently if the cells had been cultured in the absence or presence of DFMO (*Figure 4C and D*), indicating that the altered GSH and T(SH)$_2$ concentrations had no effect on the steady state thiol/disulfide ratios in the cytosol. The rate and degree of sensor oxidation triggered by exogenously applied H$_2$O$_2$ was also unaffected. However, recovery of reduced Tpx-roGFP2 appeared to be slightly delayed in cells that

had been cultured in the presence of DFMO (*Figure 4C*). Upon diamide stress, reductive recovery was clearly delayed in the drug-treated cells (*Figure 4D*). Cells that were grown with DFMO for 72 hr showed an identical behavior (*Figure 4—figure supplement 1*). In summary, parasites with diminished cellular $T(SH)_2$ content exhibit decreased reductive capacity when exposed to an oxidative insult.

## Tryparedoxin is required for cytosolic roGFP2-hGrx1 and roGFP2 to respond to $H_2O_2$

PC *T. brucei* cell lines constitutively expressing roGFP2-hGrx1 or roGFP2 were transfected with a construct that allowed the Tet-inducible depletion of Tpx. Tpx-roGFP2-expressing cells could not be analyzed since induction of RNA interference (RNAi) against Tpx also depleted the sensor (not shown). After 24 hr of RNAi, Tpx was significantly down-regulated (*Figure 5A and B*), but proliferation was not yet affected. After 72 hr the cells died in accordance with the fact that Tpx is essential (*Figure 5—figure supplement 1A*). The same behavior was previously observed for BS *T. brucei*, and attributed to a transient increase in the free thiol levels (*Comini et al., 2007*). For Tpx-depleted PC cells, we obtained 10–20% higher GSH and $T(SH)_2$ concentrations compared to the non-induced controls (*Figure 5—figure supplement 1B*).

As expected, the sensor response towards exogenous $H_2O_2$ in non-induced cells was identical to that in the parental cell line (*Figure 5C and D*, left graphs and *Figure 3D and F*). In the induced cells, the sensors still showed a basal OxD of almost zero indicating that Tpx depletion as such did not affect the cytosolic $T(SH)_2/TS_2$ steady state ratio. However, both sensors reacted only marginally when the cells were challenged with $H_2O_2$ (*Figure 5C and D*, right graphs). A direct effect on sensor sensitivity can be ruled out because only unfused roGFP2, but not roGFP2-hGrx1, requires the presence of Tpx to equilibrate with the $T(SH)_2/TS_2$ redox couple. In the absence of Tpx, the parasite

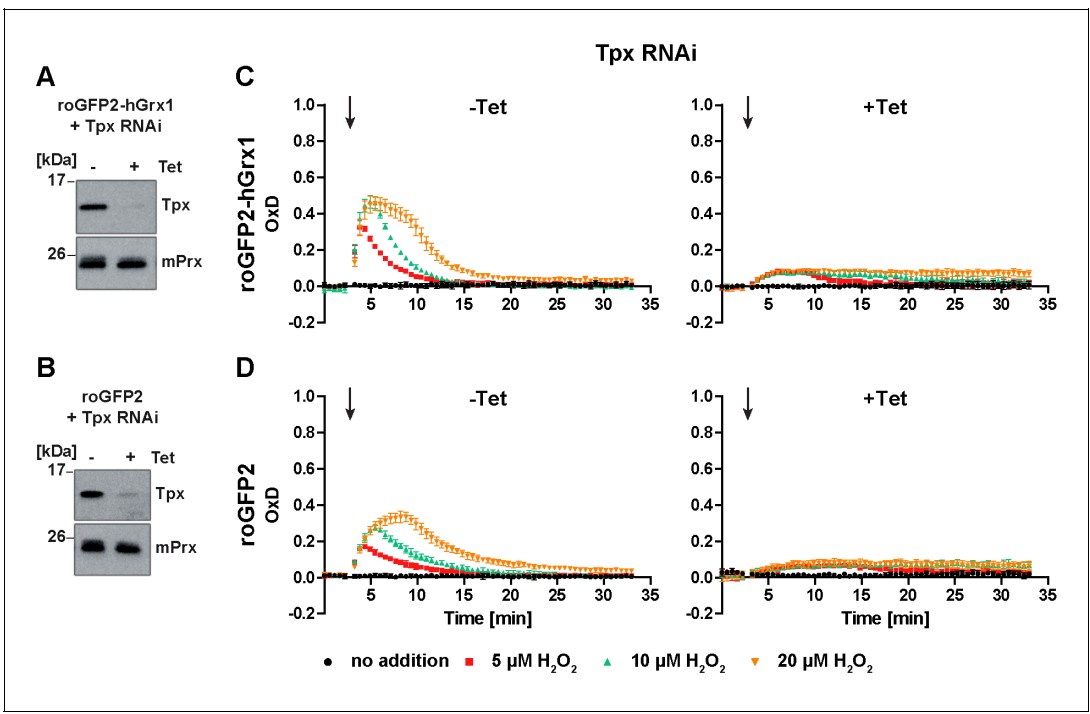

**Figure 5.** Depletion of Tpx abolishes the cytosolic sensor response to exogenous $H_2O_2$. PC cells constitutively expressing (**A, C**) roGFP2-hGrx1 or (**B, D**) roGFP2 and harboring a construct for Tet-inducible RNAi against Tpx were cultured for 24 hr in the absence (-) or presence (+) of Tet. (**A, B**) Western blot analysis of non-induced (-) and induced (+) cells, using antibodies against Tpx, and mPrx as loading control. (**C, D**) Plate reader-based fluorescence measurements. A single pulse of different $H_2O_2$ concentrations was given after 3 min (arrow). The values are the mean ± SD of three individual experiments with one clone. A second cell line showed identical results.

The online version of this article includes the following figure supplement(s) for figure 5:

**Figure supplement 1.** Growth phenotype and thiol content of Tpx-depleted PC *T. brucei*.

peroxidases are unable to reduce $H_2O_2$. Under these conditions, $H_2O_2$ can only be reduced by a direct reaction with $T(SH)_2$ (*Hillebrand et al., 2003*). The spontaneous formation of $TS_2$ in this reaction is probably too slow (and the subsequent reduction of $TS_2$ by TR too fast) to result in measurable probe oxidation.

## Depletion of lipid hydroperoxide-reducing peroxidases renders the cytosol more oxidizing

PC parasites lacking Px I-III undergo a ferroptosis-like cell death that starts at the mitochondrion. The cells are, however, fully viable and proliferative if the medium is supplemented with Trolox (*Bogacz and Krauth-Siegel, 2018*). PC *T. brucei* harboring Tpx-roGFP2 (*Figure 6A*), roGFP2-hGrx1 (*Figure 6B*) or roGFP2 (*Figure 6C*) were transfected with a construct for Tet-inducible depletion of Px I-III (*Schlecker et al., 2005*). RNAi was induced while keeping the cells in medium with Trolox. When control cells that were kept without Trolox showed a growth defect (usually between 18 and

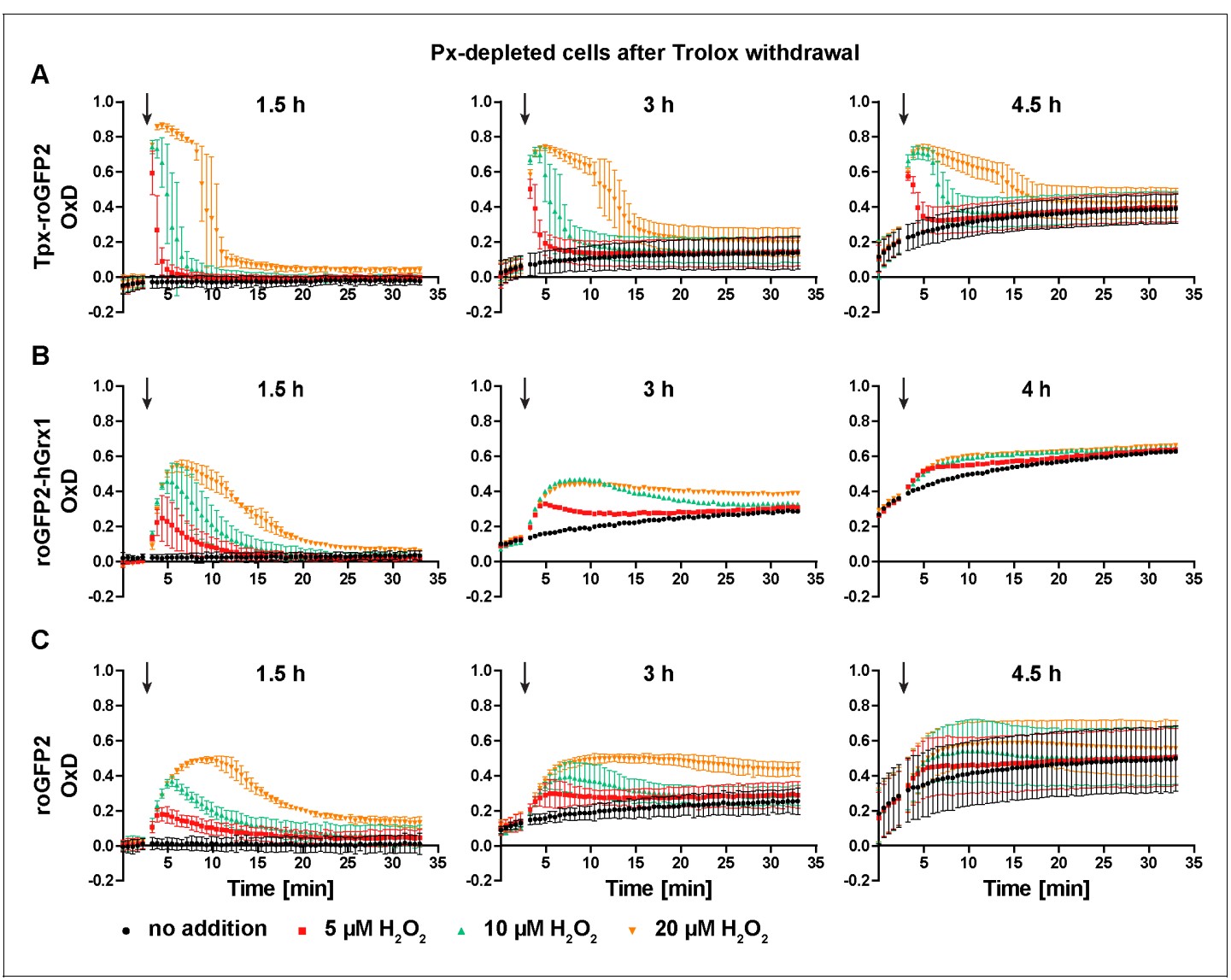

**Figure 6.** Px-depleted parasites show an increased basal sensor OxD. Parasites expressing either (**A**) Tpx-roGFP2, (**B**) roGFP2-hGrx1, or (**C**) roGFP2 were transfected with a construct that allowed the Tet-inducible depletion of the Px-type enzymes. RNAi was induced in the presence of Trolox. Subsequently the cells were transferred into Trolox-free medium and subjected to fluorescence measurements. The indicated concentrations of $H_2O_2$ were injected after 3 min (arrow). The data are the mean ± SD of three independent experiments except for the 3 hr and 4 hr measurements with the roGFP2-hGrx1 cells which were done only in duplicate and show the mean.

43 hr), the + Trolox cells were transferred into Trolox-free medium. 1.5 hr after Trolox withdrawal, all three cell lines still displayed a basal OxD of zero and responded to a bolus of $H_2O_2$ like the parental cell line, yet sensor reduction was slightly delayed (*Figure 6*, left graphs and *Figure 3B,D and F*). After 3 hr in the absence of Trolox, the basal OxD of all three sensors increased during the course of the experiment. The response to an exogenous $H_2O_2$ bolus was weaker and the sensors were only partially re-reduced when cells were treated with 20 μM $H_2O_2$ (*Figure 6*, central graphs). After keeping cells without Trolox for 4 or 4.5 hr, the sensors reported a basal OxD of 0.1–0.3 which further increased to 0.4–0.6 towards the end of the measurement (*Figure 6*, right graphs). At this time point, the cultures already contained dead cells, and only Tpx-roGFP2 still responded to an $H_2O_2$ bolus (*Figure 6A*, right graph). Taken together, in the absence of Trolox, the cytosolic probes were increasingly oxidized over time. These findings demonstrate that cells that are unable to reduce lipid hydroperoxides suffer a decrease in the cytosolic $T(SH)_2/TS_2$ ratio, associated with an impaired ability to cope with exogenously applied oxidants. Oxidation of the cytosolic trypanothione pool precedes lysis of Px-depleted cells.

## The mitochondrion harbors a trypanothione-based thiol system but is less reducing than the cytosol

To identify the so far unknown mitochondrial thiol system of trypanosomes, we generated cell lines that express roGFP2-based sensors in the mitochondrial matrix (*Figure 7—figure supplement 1*). Live cell fluorescence microscopy confirmed the specific localization of the probes in the single mitochondrion of the parasite (*Figure 7A*). The matrix-targeted mito-roGFP2-Tpx, mito-roGFP2-hGrx1, and mito-roGFP2 reported a steady state OxD of about 0.1 (*Figure 7B–G*). Unprocessed fluorescence curves are shown in *Figure 7—figure supplement 2*. These data revealed that the mitochondrion has a trypanothione-based reducing system, but with a slightly lower reducing capacity than the cytosol.

When the parasites were challenged with exogenous $H_2O_2$, the Tpx-coupled probe displayed the fastest and strongest response (*Figure 7B,D and F*). The overall sensitivity of the parasites towards $H_2O_2$ was unaffected by the presence of the sensor (*Figure 7—figure supplement 3*). All three mitochondrial sensors were oxidized to a higher degree and recovery of the reduced form was slower relative to the respective cytosolic probes (*Figure 3B,D and F*). Interestingly, unfused mito-roGFP2 was able to respond to an exogenously applied $H_2O_2$ bolus (*Figure 7F*). This strongly suggested the presence of an oxidoreductase which uses $T(SH)_2$ to reduce mitochondrial thiol peroxidases (see also next section). In contrast to the situation in the cytosol, the mitochondrial probes responded to a second 50 μM $H_2O_2$ pulse (*Figure 7B,D and F* and *Figure 3B,D and F*) and diamide treatment led to stronger oxidation compared to the cytosolic sensors (*Figure 7C,E and G* and *Figure 3C,E and G*). Again, reductive recovery was fastest for mito-roGFP2-Tpx. The mitochondrial sensors recovered from both oxidants more slowly than the respective cytosolic sensors.

To investigate the possible influence of heat on the cellular thiol redox status, PC parasites expressing the roGFP2-based sensors, either in the cytosol or mitochondrial matrix, were cultured for 24 hr at 37°C. The cells multiplied 4- to 6-times compared to the 10-times under standard culture conditions. They did not reveal any significant difference in the basal OxD values nor in the response to exogenous $H_2O_2$ when grown at either 27°C or 37°C (*Figure 7—figure supplement 4*, compare with *Figure 3B,D and F* and *Figure 7B,D and F*). The proliferation defect of the parasites at high temperature does not appear to be related to any significant change in their cytosolic or mitochondrial $T(SH)_2/TS_2$ redox state.

In vitro, lipoamide has been shown to be able to replace $T(SH)_2$ as a reductant for *L. infantum* mPrx, leading to speculations about a putative involvement of 2-oxo acid dehydrogenase complexes in the mitochondrial hydroperoxide metabolism (*Castro et al., 2008*). To further corroborate that the mitochondrial matrix harbors a trypanothione-based thiol system, cells expressing mito-roGFP2-Tpx were treated with DFMO (*Figure 7—figure supplement 5*). Inhibition of the cytosolic de novo synthesis of $T(SH)_2$ affected the recovery of the mitochondrial sensor from exogenously applied diamide (*Figure 7—figure supplement 5*). Taken together, we provide evidence in favor of the notion that the single mitochondrion of trypanosomes harbors a trypanothione-based thiol system, with a reducing capacity slightly lower than that in the cytosol.

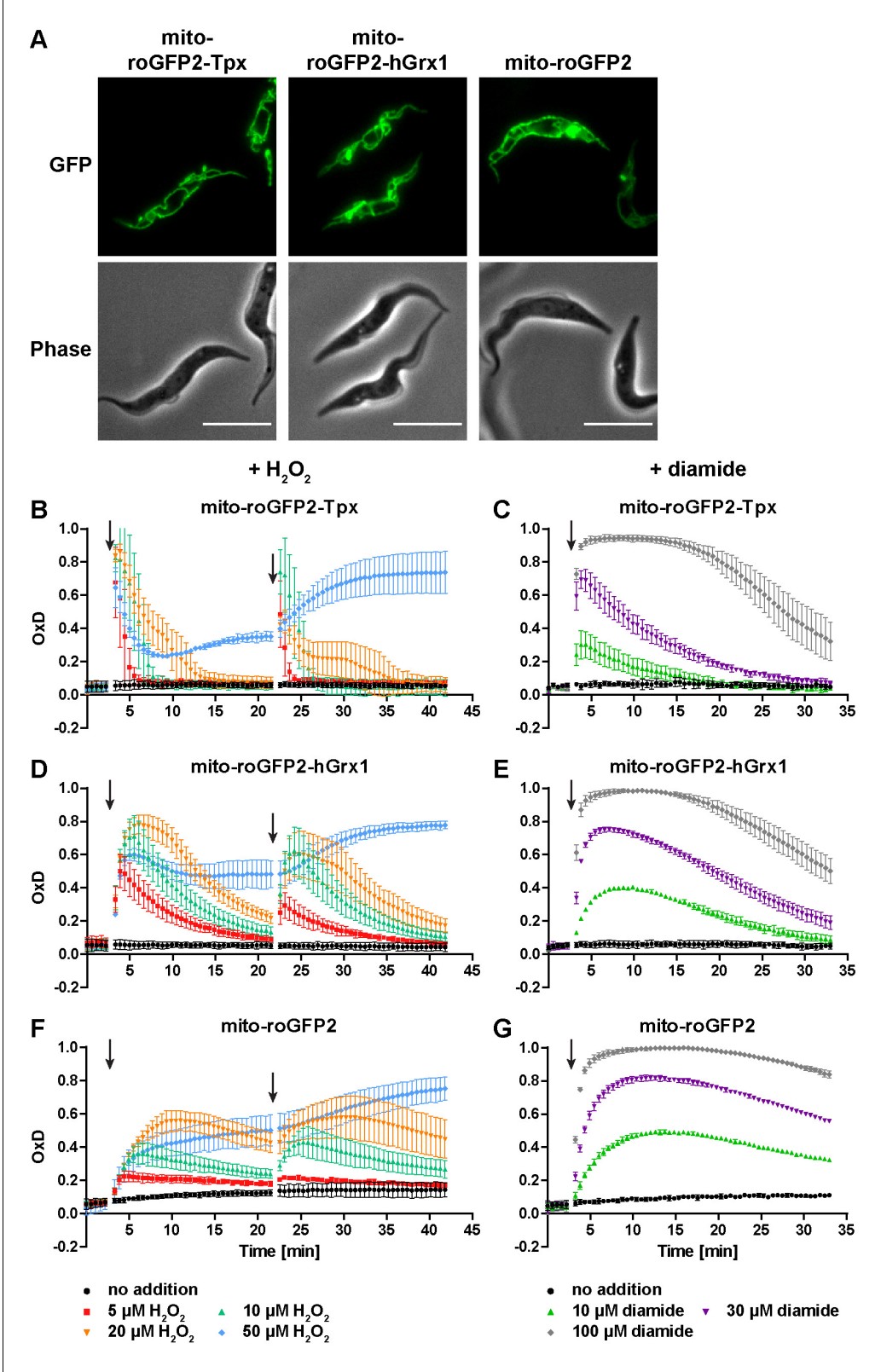

**Figure 7.** The mitochondrion has a lower reducing capacity than the cytosol. (**A**) Live cell fluorescence microscopy of PC *T. brucei* constitutively expressing mito-roGFP2-Tpx, mito-roGFP2-hGrx1, and mito-roGFP2. The GFP signal shows the mitochondrial localization of the respective sensor. Phase: phase contrast image. Scale bar: 10 μm. Cells expressing (**B, C**) mito-roGFP2-Tpx, (**D, E**) mito-roGFP2-hGrx1, or (**F, G**) mito-roGFP2 were treated with (**B, D, F**)
*Figure 7 continued on next page*

*Figure 7 continued*

$H_2O_2$ and (**C, E, G**) diamide and subjected to fluorescence measurements. The data represent the mean ± SD of at least three independent experiments.

The online version of this article includes the following source data and figure supplement(s) for figure 7:

**Figure supplement 1.** Cloning of constructs to express the redox sensors in the mitochondrial matrix of PC *T. brucei*.

**Figure supplement 2.** Unprocessed fluorescence intensity curves for cells expressing (**A**) mito-roGFP2-Tpx, (**B**) mito-roGFP2-hGrx1 and (**C**) mito-roGFP2 that were challenged twice with 10 μM $H_2O_2$ (arrows).

**Figure supplement 2—source data 1.** Original data for *Figure 7—figure supplement 2*.

**Figure supplement 3.** Sensitivity of mito-roGFP2-Tpx expressing PC *T. brucei* towards $H_2O_2$.

**Figure supplement 4.** Heat stress does not affect the thiol redox status in the cytosol or mitochondrion of PC *T. brucei*.

**Figure supplement 5.** Cultivation with DFMO decelerates the reduction rate of mito-roGFP2-Tpx.

## Tpx depletion partially affects the response of the mitochondrial sensors towards exogenous $H_2O_2$

Based on the finding that Tpx depletion abolished the ability of cytosolic roGFP2-hGrx1 and roGFP2 to respond to an exogenous $H_2O_2$ bolus, we asked if Tpx affects also the response of the mitochondrial probes. PC *T. brucei* expressing mito-roGFP2-hGrx1 or mito-roGFP2 were transfected with the construct for inducible RNAi against Tpx as described above. After 24 hr of Tet-induction, the cells showed a strong depletion of the oxidoreductase (*Figure 8A and B*), but not yet a proliferation defect.

The response of the non-induced cells towards exogenous $H_2O_2$ corresponded to that of the respective parental cell line (*Figure 8C and D*, left graphs and *Figure 7D and F*). Notably, in Tpx-

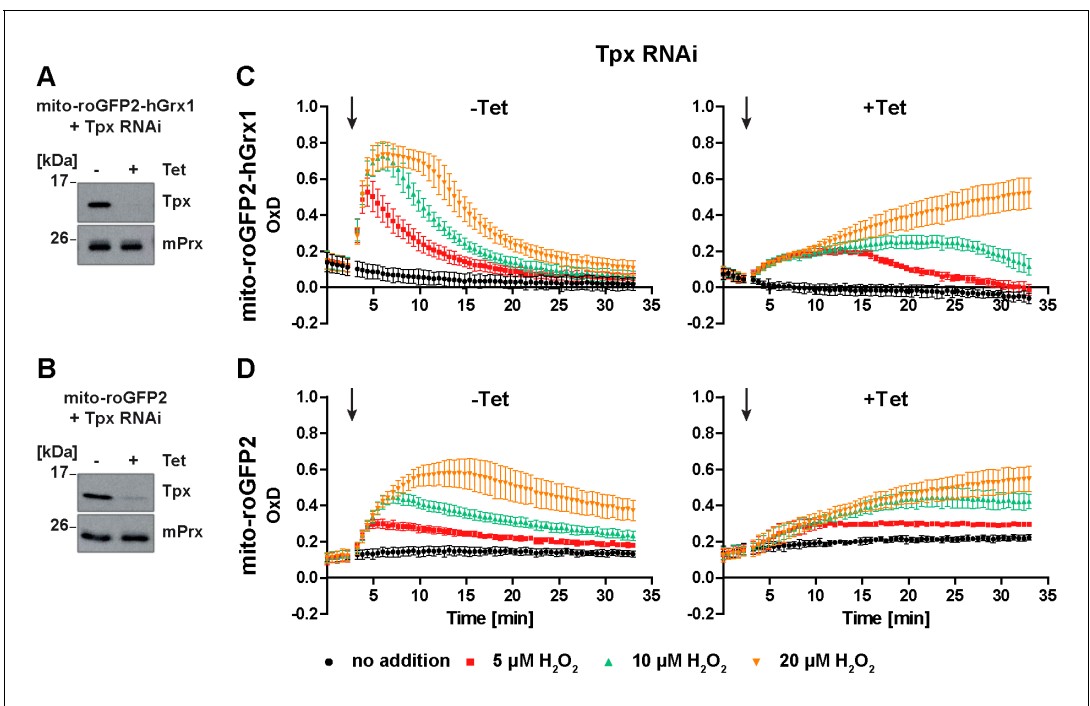

**Figure 8.** In Tpx-depleted cells the mitochondrial sensors are still able to respond to exogenous $H_2O_2$ stress. (**A, C**) Mito-roGFP2-hGrx1- and (**B, D**) mito-roGFP2-expressing Tpx RNAi cell lines were cultured for 24 hr in the absence (-) or presence (+) of Tet. (**A, B**) Total lysates of $2 \times 10^6$ of non-induced (-) and induced (+) cells were subjected to Western blot analysis using antibodies against Tpx and mPrx as loading control. (**C, D**) The cells were subjected to plate reader-based fluorescence measurements. After 3 min, a single pulse of different $H_2O_2$ concentrations was injected (arrow). (**C**) The values are the mean ± SD of at least two individual experiments using three different clones. (**D**) The data represent the mean ± SD of three independent experiments with one clone. A second cell line gave identical results.

depleted cells, both mito-roGFP2-hGrx1 and mito-roGFP2 were still able to respond to an exogenous $H_2O_2$ bolus (*Figure 8C and D*, right graphs). However, oxidation of mito-roGFP2-hGrx1 by 5 µM or 10 µM $H_2O_2$ was slow and weak, and re-reduction of the sensor was delayed as compared to non-induced cells. When 20 µM $H_2O_2$ was applied, sensor oxidation was still increasing at the end of the experiment (*Figure 8C*, right graph). In the case of mito-roGFP2, oxidation by 10 µM and 20 µM $H_2O_2$ did not reach a maximum within the observation time (*Figure 8D*, right graph). Taken together, in cells lacking Tpx, the mitochondrial sensors are still able to respond to exogenously applied $H_2O_2$, in sharp contrast to the situation in the cytosol (*Figure 5C and D*, right graphs). Nevertheless, the response was significantly weaker than in cells proficient in Tpx. Based on these data we hypothesize that a small portion of cytosolic Tpx may be present in the mitochondrial matrix, as suggested previously by immune-gold labeling (*Tetaud et al., 2001*), and that the mitochondrion harbors another oxidoreductase that is able to transfer electrons between the trypanothione redox couple and mitochondrial thiol peroxidases.

### The mitochondrial peroxiredoxin contributes to the reduction of exogenously applied $H_2O_2$ and its depletion affects the cytosolic sensor response

Depletion of mPrx in PC *T. brucei* over three days results in undetectable levels of the protein. At this time point, the cells still proliferate like WT cells, whereas after seven days of depletion, proliferation is severely affected and the parasites display a highly elongated shape (Bogacz et al. unpublished data). Cells expressing Tpx-roGFP2 or uncoupled roGFP2 in either the mitochondrial matrix or the cytosol were transfected with a construct for Tet-inducible depletion of mPrx. RNAi was induced for three or seven days. The altered morphology of the mPrx-depleted cells had no effect on the expression level and subcellular targeting of the sensors (*Figure 9A*). Non-induced cells displayed the same sensor response as the respective parental strain (*Figure 9B–E*, left graphs compare with *Figure 7B,D and F* and *Figure 3B,D and F*).

In mPrx-depleted cells, the basal OxD of all sensors was unchanged (*Figure 9B–E*) indicating that neither the mitochondrial nor the cytosolic T(SH)$_2$/TS$_2$ ratio was affected. However, depletion of mPrx had a strong impact on the sensor response when the cells were exposed to exogenous $H_2O_2$. In cells subjected to RNAi for three days, the mitochondrial sensors showed virtually no oxidation when 5 µM or 10 µM $H_2O_2$ was applied. Addition of 20 µM $H_2O_2$ resulted in a slow and minor increase in OxD, together with a delayed reductive recovery (*Figure 9B and C*, middle graphs). After seven days of mPrx depletion, oxidation of the mitochondrial sensors was again increased and re-reduction even more delayed (*Figure 9B and C*, right graphs). Interestingly, RNAi against mPrx had also a strong effect on the response of the cytosolic sensors. In cells subjected to mPrx depletion for three days and exposed to a bolus of 20 µM $H_2O_2$, re-reduction of cytosolic Tpx-roGFP2 was slowed down and roGFP2 showed a slightly increased OxD peak (*Figure 9D and E*, middle graphs). In cells induced for seven days and treated with 20 µM $H_2O_2$, Tpx-roGFP2 displayed an OxD peak lasting for at least 10 min. At all $H_2O_2$ concentrations used, re-reduction of cytosolic Tpx-roGFP2 was markedly attenuated relative to non-induced cells (*Figure 9D*, right and left graphs). At this time point, also roGFP2 was oxidized to a higher degree and for a longer time (*Figure 9E*, right graph). Thus, depletion of mPrx affected the response of mito-roGFP2-Tpx and mito-roGFP2 to a challenge with exogenous $H_2O_2$. This indicates that mPrx is involved in the reduction of exogenously applied $H_2O_2$ and is the main mitochondrial $H_2O_2$-metabolising thiol peroxidase. However, the mitochondrial sensor response was not completely lost. Probably, $H_2O_2$ can still be reduced by Px III (*Bogacz and Krauth-Siegel, 2018*; *Diechtierow and Krauth-Siegel, 2011*; *Schaffroth et al., 2016*). In cells that were subjected to long-term mPrx depletion, reduction of exogenously applied $H_2O_2$ in the cytosol was strongly delayed. One may speculate that this is due to a general impairment of metabolism in these proliferation arrested cells.

## Discussion

The expression of Tpx-roGFP2, a novel biosensor, in African trypanosomes allowed for the first time to follow real-time changes of the trypanothione redox state and, most importantly, to demonstrate that the parasites harbor a mitochondrial T(SH)$_2$/TS$_2$ system. Remarkably, hGrx1-roGFP2 also equilibrated with the parasite thiol/disulfide couples, albeit less efficiently than the Tpx-roGFP2 sensor. As

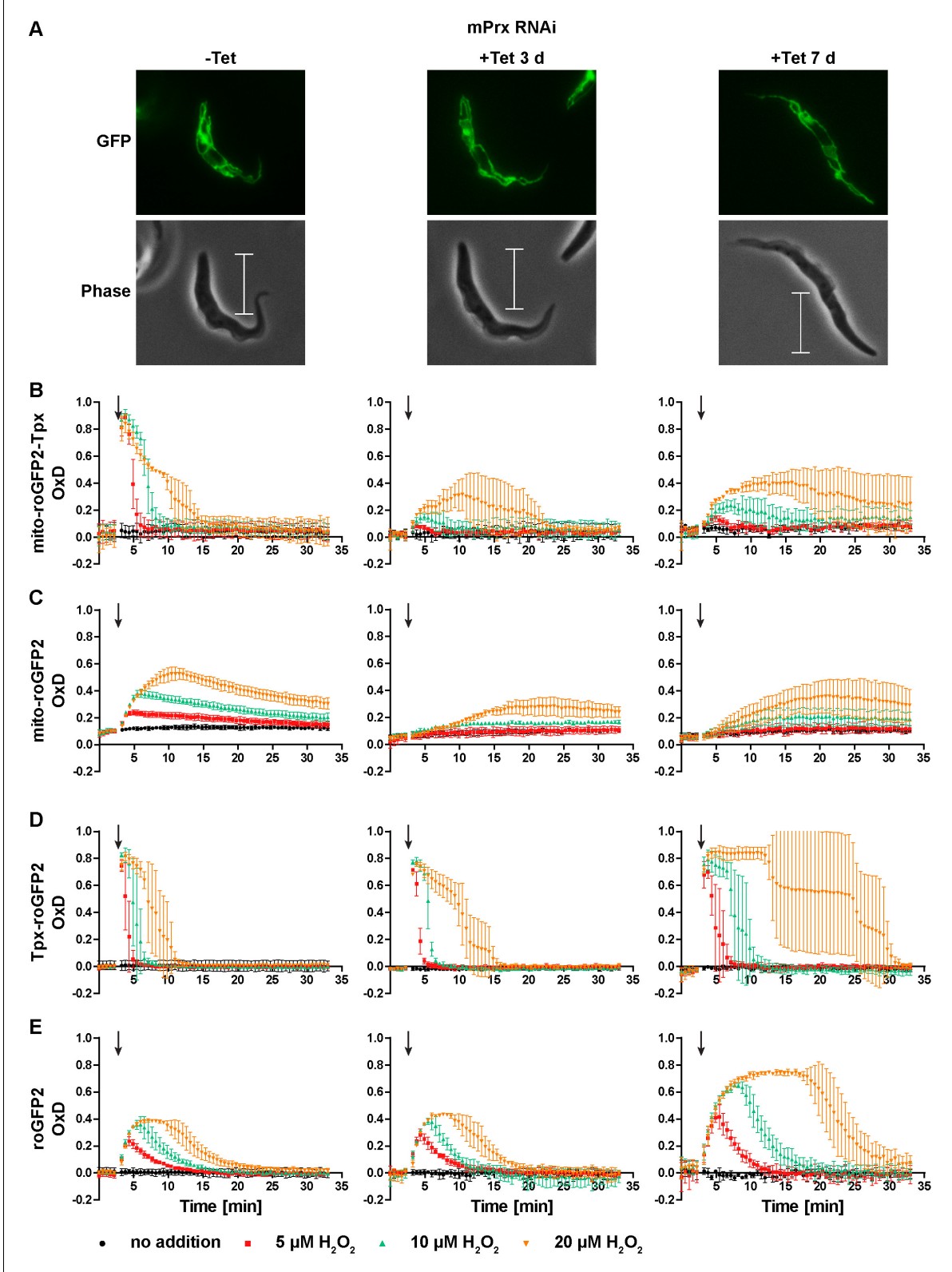

**Figure 9.** Depletion of mPrx influences both mitochondrial and cytosolic sensor responses. (A) Live cell imaging of mito-roGFP2-Tpx expressing PC *T. brucei* that harbor a construct for Tet-inducible RNAi against mPrx. The GFP signal confirmed the mitochondrial localization of the sensor under all conditions and for all time points. The phase contrast images show the highly elongated shape the cells adapt upon mPrx depletion. Scale bar: 10 μm. mPrx RNAi cells expressing the (B, D) Tpx-coupled or (C, E) unfused sensor in the (B–C) mitochondrial matrix or in the (D–E) cytosol were cultured for

*Figure 9 continued on next page*

Figure 9 continued

seven days in the absence (left) and for three (middle) and seven days (right) in the presence of Tet. Cells were subjected to fluorescence plate reader-based measurements. A single pulse of different $H_2O_2$ concentrations was injected after 3 min (arrow). The values are the mean ± SD of three individual experiments with one clone. A second cell line showed identical results.

expected, unfused roGFP2 showed an overall low reactivity. Nevertheless, $T(SH)_2$ was the most efficient reductant for all three sensors. This finding does not appear to be related to the redox potential as the $E_0'$ of $T(SH)_2$ has been reported to be only slightly more negative (−242 mV) than that of GSH (−230 mV) (*Fairlamb and Cerami, 1992*). However, formation of an intramolecular disulfide ($TS_2$) can be expected to be kinetically more favorable than the formation of intermolecular disulfides (GSSG, $Gsp_2$). Another crucial point is that the thiol pK-value of $T(SH)_2$ is at least one pH unit lower than that of GSH. Thus, at physiological pH values, a significantly higher portion of $T(SH)_2$ is present in the reactive deprotonated form (*Manta et al., 2013*; *Moutiez et al., 1994*).

When treated with small increments of $TS_2$ or GSSG in the presence of cellular GSH and $T(SH)_2$ concentrations, both Tpx-roGFP2 and hGrx1-roGFP2 were oxidized, with $TS_2$ yielding a slightly faster response. As trypanosomes lack glutathione-dependent peroxidases and GSH is regenerated from GSSG by thiol-disulfide exchange with $T(SH)_2$, an in vivo situation that could lead to the specific formation of GSSG is not known. In the physiological context of trypanosomes, both Tpx-roGFP2 and hGrx1-roGFP2 equilibrate with the $T(SH)_2$/$TS_2$ redox couple. The reactivity of Grxs towards $T(SH)_2$ has been observed in previous studies showing that the parasite dithiol reduces *E. coli* Grx1 as efficiently as dithioerythritol (*Ceylan et al., 2010*).

Tpx-roGFP2, roGFP2-hGrx1 and roGFP2 expressed in PC trypanosomes reported a fully reduced cytosol, similar to other organisms (*Gutscher et al., 2008*; *Kasozi et al., 2013*; *Loi et al., 2017*; *Morgan et al., 2013*) and the reduced state was rapidly restored after a challenge with $H_2O_2$. The sensor response to exogenous $H_2O_2$ is most likely mediated by cellular thiol peroxidases (*Meyer and Dick, 2010*). The parasite enzymes use the $T(SH)_2$/Tpx system for re-reduction and thus enzymatically generate $TS_2$ (*Castro and Tomás, 2008*; *Krauth-Siegel and Comini, 2008*) which is then reported by the biosensors. Tpx-roGFP2 showed the fastest and strongest response, but also roGFP2-hGrx1 and even unfused roGFP2 reversibly responded to exogenous oxidants. This strongly suggested that all three sensors reacted to changes of the $T(SH)_2$/$TS_2$ ratio in vivo. BS *T. brucei* expressing hGrx1-roGFP2 in the cytosol show sensor oxidation in response to treatment with different anti-parasitic compounds which was proposed to reflect a decrease in the GSH/GSSG ratio (*Franco et al., 2017b*; *Franco et al., 2017a*). However, as shown here, it is more likely that the compounds affected the cytosolic $T(SH)_2$/$TS_2$ redox state.

When PC cells were treated with diamide, Tpx-roGFP2 displayed a lower maximum OxD and shorter lag phase compared to roGFP2-hGrx1 and roGFP2, in accordance with the efficient in vitro reduction of the sensor by low micromolar concentrations of $T(SH)_2$. In BS *T. brucei*, $H_2O_2$-treatment results primarily in the formation of free disulfides, whereas in the case of diamide, the main products are protein-bound thiols (*Ulrich et al., 2017*). The generally slower probe re-reduction after diamide- relative to $H_2O_2$-treatment may be due to a slower reduction of protein-mixed disulfides than of $TS_2$ by TR.

In *Staphylococcus aureus* and *Corynebacterium glutamicum*, depletion of bacillithiol and mycothiol, respectively, leads to basal sensor oxidation (*Loi et al., 2017*; *Tung et al., 2019*). When PC *T. brucei* were grown in the presence of DFMO, the cellular GSH was almost doubled and $T(SH)_2$ dropped to 40% of the concentration in untreated cells. Nevertheless, the basal OxD of Tpx-roGFP2 remained close to zero. This is consistent with data from BS trypanosomes, where depletion of TryS also causes an increase of GSH and a decrease of $T(SH)_2$ (*Comini et al., 2004*) without affecting the cellular thiol-disulfide ratio (*Ulrich et al., 2017*). Notably, in the DFMO-treated cells, the total thiol concentration (GSH plus 2 x $T(SH)_2$) was lowered by 63%. This further corroborates that the cytosolic redox potential is unaffected by these changes in thiol concentration. Indeed, the total acid-soluble thiol content can vary by up to 20-fold between life cycle stages, yet intact trypanosomes and *Leishmania* differ only 2-fold in their capacity to metabolize $H_2O_2$ between stages (*Ariyanayagam and Fairlamb, 2001*). In the DFMO-treated cells, recovery of reduced Tpx-roGFP2 after an exogenous oxidative challenge was slowed down whereby the effect was less pronounced with $H_2O_2$ compared

to diamide. Thus, the cytosolic peroxidases were still able to efficiently reduce $H_2O_2$, even at the significantly lower $T(SH)_2$ level.

In an *Arabidopsis thaliana* mutant with 20% GSH compared to WT cells, the cellular GSH/GSSG ratio is unaffected. However, cytosolic roGFP2 is slightly oxidized because the lowered glutathione level shifts the glutathione redox potential to less negative values (*Meyer et al., 2007*). The redox state of Tpx-roGFP2 was unaffected in the DFMO-treated parasites. Oxidation of $T(SH)_2$ results in the formation of an intramolecular disulfide. This distinguishes trypanosomatids from all other organisms. Independently of the nature of the individual thiol, mammals, yeast, plants and bacteria employ a monothiol for regulating the cellular redox homeostasis which renders the redox potential sensitive to changes in the overall thiol concentration (*Loi et al., 2017*; *Meyer et al., 2007*; *Tung et al., 2019*). In the case of the $T(SH)_2/TS_2$ couple, the redox potential and corresponding steady state sensor response are only determined by the thiol-disulfide ratio.

*P. falciparum* strains with elevated GSH levels display a lower sensor OxD (*Kasozi et al., 2013*) reflecting an increased reducing capacity. Depletion of Tpx resulted in a 10–20% increase of the intracellular GSH and $T(SH)_2$ levels in PC *T. brucei*. As the steady state OxD of the cytosolic sensors was almost zero already in the non-induced cells, a further decrease in the Tpx-depleted cells could not be observed and was also not expected. In the absence of Tpx, both roGFP2 and roGFP2-hGrx1 did not respond to exogenous $H_2O_2$. Tpx depletion abolishes the electron transfer from $T(SH)_2$ onto the parasite peroxidases and thus the enzymatic production of $TS_2$. This provided additional evidence that in trypanosomes, roGFP2-hGrx1 senses changes in the $T(SH)_2/TS_2$ system. In cells in which the Px-type peroxidases were depleted, all three biosensors showed a dramatic increase of the steady state OxD. Thus, lipid peroxidation rapidly translates into the oxidation of the cytosolic trypanothione pool.

When targeted to the mitochondrion, the redox sensors revealed an overall similar response as the cytosolic probes. This was strong evidence that the mitochondrial matrix harbors a trypanothione-based thiol metabolism. However, the slightly elevated basal OxD of the mitochondrial probes indicated that the mitochondrion is less reducing than the cytosol, in accordance with data reported for yeast cells (*Kojer et al., 2012*). When the parasites were exposed to exogenous oxidants, all three mitochondrial probes were oxidized to a higher degree and were more slowly re-reduced than the respective cytosolic sensors. Also in yeast cells, the matrix glutathione pool was found to be significantly more sensitive to $H_2O_2$-induced oxidation than the cytosolic glutathione pool even though the $H_2O_2$ concentration in the matrix is lower than in the cytosol (*Calabrese et al., 2019*). This indicates that the reducing capacity in the single mitochondrion of trypanosomes is comparably low which could be due to a diminished $T(SH)_2/TS_2$ ratio and/or low total thiol concentration. In contrast to the cytosolic sensors, the mitochondrial probes responded to a second pulse of 50 µM $H_2O_2$. This suggested that the mPrx is less prone to over-oxidation than the cPrx as shown for mammalian Prx3 (*Cox et al., 2010*); or less $H_2O_2$ reaches the mitochondrion due to the buffering capacity of the cytosol (*Calabrese et al., 2019*; *Morgan et al., 2016*). To analyze this point in more detail, future studies should include probes that directly record changes in the $H_2O_2$ concentration (*Morgan et al., 2016*).

The de novo synthesis of $T(SH)_2$ and the reduction of $TS_2$ are confined to the cytosol (*Oza et al., 2005*; *Smith et al., 1991*). Therefore, the question arises how $T(SH)_2$ reaches the mitochondrial matrix and is then kept in the reduced state. So far, specific transporters for $T(SH)_2/TS_2$ have not been identified. In yeast and human cells, the biosynthesis of GSH is also cytosolic, but GR occurs in both the cytosol and mitochondrial matrix (*Calabrese et al., 2017*). In rabbit kidney mitochondria and mitoplasts, the dicarboxylate carrier (DIC) and oxoglutarate carrier (OGC) were identified as GSH transporters in the inner mitochondrial membrane (*Chen et al., 2000*), whereas transport of GSH in *Lactococcus lactis* is independent of the two anion carriers (*Booty et al., 2015*). In *T. brucei*, the mitochondrial carrier protein 12 (MCP12) was shown to act as carboxylate and tricarboxylate carrier, thereby fulfilling the functions of DIC and OGC (*Colasante et al., 2009*; *Colasante et al., 2018*). Due to its overall positive charge, it is highly unlikely that $T(SH)_2$ is imported by these transporters. Regeneration of reduced mitochondrial hGrx1-roGFP2 in oxidatively challenged yeast cells requires the matrix-localized GR (*Kojer et al., 2012*). This is not the case in trypanosomes. The basal OxD of the sensors in the matrix was only slightly higher than in the cytosol and the probes reversibly responded to exogenous oxidative challenges. One may thus speculate that trypanosomes possess a protein that mediates an exchange of $TS_2$ and $T(SH)_2$ over the inner mitochondrial membrane.

The finding that unfused mito-roGFP2 responded to exogenous $H_2O_2$ pointed to the presence of an oxidoreductase that facilitates the equilibration of the sensor with the mitochondrial $T(SH)_2/TS_2$ couple and is capable of transferring reducing equivalents from $T(SH)_2$ onto the mitochondrial peroxidases. Nevertheless, re-reduction of mito-roGFP2 was slow when compared to cytosolic roGFP2. The catalytic capacity of the mitochondrial oxidoreductase appears to be limited due to either low reactivity or low concentration. Tpx-depletion affected the response of mito-roGFP2 and mito-roGFP2-hGrx1. One putative explanation may be that a minor fraction of Tpx resides in the mitochondrion. This would be in agreement with immuno-electron microscopy data which revealed some gold particles in the mitochondrion (*Tetaud et al., 2001*) although immunofluorescence studies and fractionated cell lysis displayed a cytosolic localization of Tpx (*Currier et al., 2019*; *Ebersoll et al., 2018*; *Tetaud et al., 2001*). Notably, Tpx-depletion did not fully abolish the response of the mitochondrial sensors to exogenous $H_2O_2$. Probably, another oxidoreductase can mediate the electron transfer between $T(SH)_2$ and the mitochondrial peroxidases. With respect to the identity of this protein we can only speculate. A second *T. brucei* Tpx is located in the outer mitochondrial membrane facing the cytosol (*Castro et al., 2010*). Other proteins that might catalyze the electron transfer from $T(SH)_2$ are Grxs and Trxs. *T. brucei* Grx1 is a cytosolic protein. Grx2 is located in the IMS and is unable to replace Tpx in the Px-catalyzed peroxidase assay (*Ebersoll et al., 2018*; *Ceylan et al., 2010*). *T. brucei* Trx1 lacks a mitochondrial pre-sequence, but a genome-wide localization study revealed the N-terminally mNeonGreen-tagged protein in the mitochondrion (*Dean et al., 2017*). Trx1 reduces Px and cPrx albeit with much lower efficiency than Tpx (*Hillebrand et al., 2003*; *Schmidt and Krauth-Siegel, 2003*) and was inactive in an mPrx assay (unpublished data). A recently characterized mitochondrial Trx2 lacks reductase activity (*Currier et al., 2019*). In addition, human Trx1 does not engage in thiol-disulfide exchange with roGFP2, not even when genetically fused to the redox sensor (*Gutscher et al., 2008*; *Meyer and Dick, 2010*). An importome study identified 21 new mitochondrial proteins with annotated oxidoreductase activity (*Peikert et al., 2017*). Only three proteins appear to be possible candidates for the transfer of electrons from the trypanothione redox couple to the mitochondrial tryparedoxin peroxidases. These are Trx1, Grx2 and a putative glutathione-S-transferase/glutaredoxin (Tb927.7.3500). The latter protein harbors a mitochondrial targeting sequence and a CxxC motif but has not yet been characterized.

Depletion of mPrx affected neither the mitochondrial nor the cytosolic $T(SH)_2/TS_2$ steady state, similar to various peroxidase deletion mutants of *C. glutamicum* which also show no alteration in the basal mycothiol redox potential (*Tung et al., 2019*). After three days of RNAi against mPrx, the protein was strongly depleted but the cells still proliferated like WT parasites. When these cells were treated with $H_2O_2$, mitochondrial roGFP2-Tpx and roGFP2 displayed a markedly diminished response. Thus, in PC *T. brucei*, mPrx is the key peroxidase that couples $H_2O_2$ reduction to trypanothione oxidation in the mitochondrial matrix. This contrasts with the situation in promastigote *Leishmania* where the enzyme neither plays a role for proliferation nor for detoxification of $H_2O_2$ (*Castro et al., 2011*; *Teixeira et al., 2015*), but compares with yeast cells where $H_2O_2$-induced matrix glutathione oxidation is dependent on the presence of the mitochondrial 1-Cys-peroxiredoxin (*Calabrese et al., 2019*). After seven days of mPrx-depletion, the parasites adapted a highly elongated shape and virtually stopped proliferation. At this time point, the cytosolic sensors displayed an extremely long lag phase before they recovered from $H_2O_2$-induced oxidation. Apparently, the capacity of the cytosol to cope with exogenous oxidants was severely hampered in these growth-arrested cells, in agreement with the notion that there is a cross-talk between the two compartments. Future work should focus on the identification of the proteins that allow the transport/exchange of $T(SH)_2$ and $TS_2$ in and out of the mitochondrion.

## Materials and methods

The key resources table is provided in the *Supplementary file 1*. The primers used in this work are given in the *Supplementary file 2*.

### Materials

Cystatin, diamide, GSH, hemin, penicillin/streptomycin, pepstatin, phleomycin, phenylmethylsulfonyl fluoride (PMSF), Tet, and Trolox were from Sigma-Aldrich, Munich, Germany. Geneticin disulfate (G418), hygromycin B, and NADPH were purchased from Carl Roth, Karlsruhe, Germany and GSSG

from Serva, Heidelberg, Germany. DFMO was obtained from Cayman Chemicals, Ann Arbour, Michigan, dithiothreitol (DTT) from Biomol, Hamburg, Germany, fetal calf serum (FCS) from Biochrome, Berlin, Germany and DNaseI, $H_2O_2$, lysozyme, and mBBr from Merck Millipore, Darmstadt, Germany. Isopropyl-β-D-thiogalacto-pyranoside (IPTG) was ordered from Peqlab, Erlangen, Germany and Bond Breaker tris(2-carboxyethyl)phosphine (TCEP) solution from Thermo Scientific, Schwerte, Germany. T(SH)$_2$ and Gsp as well as TS$_2$ and Gsp$_2$ (*Comini et al., 2009*; *Leroux et al., 2013*), tag-free recombinant *T. brucei* TR (*Persch et al., 2014*), and human GR (*Nordhoff et al., 1993*) were prepared as described. Polyclonal rabbit antibodies against *T. brucei* Tpx (*Schlecker et al., 2005*) and guinea pig antibodies against mPrx (*Ebersoll et al., 2018*) were generated previously. HRP-conjugated goat antibodies against rabbit IgGs were from Thermo Scientific, Heidelberg, Germany, and donkey antibodies against guinea pig IgGs from Merck. The pQE-60_*rogfp2* (Addgene #65046), pQE-60_*hgrx1-rogfp2* (Addgene #64799; *Gutscher et al., 2008*), and pCaSpeR4_*mito-rogfp2-hgrx1* (Addgene #65000; *Albrecht et al., 2014*) plasmids were obtained as described. The pHB 136.1 TbTXNH6 (*Budde et al., 2003*) and pHD vectors were kindly provided by Drs Leopold Flohé, Montevideo, Uruguay, and Christine Clayton, Heidelberg, Germany, respectively.

## Methods

### Cloning of constructs for roGFP2 sensor expression and Tpx RNA interference

To generate the recombinant Tpx-roGFP2 sensor, the 5' *Nco*I site of *hgrx1* in the pQE-60_*hgrx1-rogfp2* vector was changed into an *Acc65*I restriction site using the QuikChange II Site-Directed Mutagenesis Kit (Agilent Technologies, Waldbronn, Germany) and the primers GroGFP-Acc-fw and -rv. The 3' *Spe*I site between *hgrx1* and the linker sequence was mutated to a *Mlu*I site (primer pair GroGFP2-Mlu-fw and -rv). The coding region of *T. brucei tpx* (Tb927.3.3760) with 5' *Acc65*I and 3' *Mlu*I restriction sites was amplified from the pHB 136.1 TbTXNH6 vector using Tpx-Acc-fw and Tpx-Mlu-rv as primers and ligated into the digested pQE-60_*hgrx1-rogfp2* vector. In the resulting pQE-60_*tpx-rogfp2* plasmid two *BamH*I restriction sites in the *tpx* coding region were removed by single point mutations (Tpx-Bam1/2-fw and -rv).

For Tet-inducible expression of the sensors in the cytosol of the parasites, the *hgrx1-rogfp2* and *tpx-rogfp2* sequences were amplified by PCR using a previously generated pHD1700_*hgrx1-rogfp2-2myc* (primer pair pHD-Grx1-fw and pHD-roGFP2-rv) and pQE-60_*tpx-rogfp2* (primers pHD-Tpx-fw and pHD-roGFP2-rv) as templates. As sequencing revealed that the pHD1700 vector contained a deletion in the resistance cassette, the amplicons were introduced into another pHD1700 plasmid through the 5' *Hind*III and 3' *BamH*I restriction sites.

For constitutive cytosolic sensor expression, the *rogfp2* sequence was amplified from the respective pQE-60 plasmid with 5' *Hind*III and 3' *BamH*I sites using the primers pHD-roGFP2-fw and pHD-roGFP2-rv and ligated into the pHD1991 vector. Using the same restriction sites, the *hgrx1-rogfp2* and *tpx-rogfp2* sequences were cut out from pHD1700_*hgrx1-rogfp2* and pHD1700_*tpx-rogfp2* and inserted into the pHD1991 vector, resulting in pHD1991_*hgrx1-rogfp2* and pHD1991_*tpx-rogfp2*. Since expression of hGrx1-roGFP2 was not successful in the parasites, a pHD1991_*rogfp2-hgrx1* plasmid with inverted domain order was generated. With the mito-roGFP2-hGrx1 plasmid (see below) as template, *rogfp2-hgrx1* without mitochondrial targeting sequence was amplified (pHD-roGFP2-fw and roGFP2-hGrx1-BamHI-rv) and ligated in the pHD1991 vector using the 5' *Hind*III and 3' *BamH*I restriction sites.

To constitutively express the sensors in the mitochondrion of the parasite, the pCaSpeR4_*mito-rogfp2-hgrx1* plasmid served as template (*Albrecht et al., 2014*). The *mito-rogfp2-hgrx1* sequence was amplified via the primers mito-HindIII-fw and roGFP2-hGrx1-BamHI-rv and firstly ligated into the pGEM-T vector (Promega, Mannheim, Germany). The internal *BamH*I site between the mitochondrial targeting sequence and *rogfp2* was deleted by mutagenesis PCR (mito-BamHI-fw and -rv). The mutated *mito-rogfp2-hgrx1* sequence was stepwise released by digestion with *Hind*III and *BamH*I and ligated into the pHD1991 plasmid, yielding pHD1991_*mito-rogfp2-hgrx1*. Subsequently, the *hgrx1* sequence was replaced by *tpx*. For this purpose, *tpx* was amplified by PCR from pHD1991_*tpx-rogfp2* using the primers roGFP2-Tpx-EcoRI-fw and roGFP2-Tpx-BamHI-rv and ligated with the *EcoR*I/*BamH*I digested pHD1991_*mito-rogfp2-hgrx1* vector, resulting in pHD1991_*mito-rogfp2-tpx*. The *mito-rogfp2* insert was obtained by PCR with mito-HindIII-fw and pHD-roGFP2-rv as

primers from the mutated pGEM-T_*mito-rogfp2-hgrx1* construct and ligated with the *Hind*III/*Bam*HI digested pHD1991 vector yielding pHD1991_*mito-roGFP2*.

The pHD678 plasmid contains a hygromycin resistance gene and allows Tet-inducible RNAi against the inserted target transcript. A 349 bp stretch of the coding sequence of *tpx* was amplified with *Hpa*I and *Eco*RI restriction sites from genomic DNA of PC *T. brucei* by PCR with the primers Tpx-RNAi-fw_2 and Tpx-RNAi-rv_2. The same stretch plus additional 55 bp with *Hind*III and *Eco*RI sites was amplified using the primers Tpx-RNAi-fw_1 and Tpx-RNAi-rv_1. Both fragments were digested with the respective restriction enzymes and ligated with the *Hind*III- and *Hpa*I-digested pHD678_*pxIII* vector (*Schlecker et al., 2005*) resulting in pHD678_*tpx*.

## Purification of the recombinant biosensors

Three liter (6 × 500 ml) cultures of recombinant BL21 (DE3) *E. coli* in 2 x YT medium with 100 µg/ml carbenicillin were grown overnight at 37°C and 180 rpm. At an $OD_{600}$ of about 0.5–0.6, expression of *rogfp2*, *hgrx1-rogfp2*, or *tpx-rogfp2* was induced by 200 µM IPTG. After overnight cultivation at 18°C, the bacteria were harvested by centrifugation, resuspended in 15 ml buffer A (50 mM sodium phosphate, 300 mM NaCl, pH 7.5) per liter of culture, containing 50 µM PMSF, 150 nM pepstatin, 4 nM cystatin, 1.5 mg lysozyme and DNase, and disintegrated by sonication. After centrifugation, the supernatant was kept on ice and the extraction repeated. The combined supernatants from the 3 l culture were loaded onto an 8 ml Ni-NTA Superflow matrix (Qiagen, Venlo, Netherlands) equilibrated in buffer A. The column was washed with 48 ml buffer A containing 10 mM imidazole, 32 ml buffer A with 45 mM imidazole, and a 40 ml gradient to 90 mM imidazole. The proteins were eluted with 125 mM imidazole in buffer A. Their purity was assessed by SDS-PAGE and Coomassie staining. Fractions containing pure protein were combined, the buffer was exchanged to 100 mM Tris, 200 mM NaCl, 50 µM EDTA, pH 7.5 and the proteins were stored at −80°C.

## Cultivation of *T. brucei*

The parasites used in this work are PC *T. brucei brucei* of the 449 cell line, a descendant of the Lister strain 427 (*Cunningham and Vickerman, 1962*). These cells stably express the Tet-repressor (*Biebinger et al., 1997*). The cells were cultured in MEM-Pros medium supplemented with 10% (v/v) heat-inactivated FCS, 7.5 µg/ml hemin, 50 U/ml penicillin, 50 µg/ml streptomycin, and 0.5 µg/ml phleomycin at 27°C. Cells constitutively expressing the sensors were kept in the presence of 30 µg/ml G418. Cell lines harboring an inducible RNAi construct were grown in the presence of 50 µg/ml hygromycin. RNAi was induced by addition of 1 µg/ml Tet. Proliferation was monitored starting with $5 \times 10^5$ cells/ml and diluting the cultures to the initial density every 24 hr. For experiments with Px-depleted cells, the cultures contained 100 µM Trolox. The effect of Px-depletion (see next section) on the sensor response was analysed after removal of the antioxidant. To study the effect of DFMO, a stock solution of 10 mg/ml DFMO was prepared in PBS. $2–3 \times 10^5$ cells/ml were treated with 5 mM DFMO and proliferation was followed. Living cells were counted in a Neubauer chamber.

## Generation of redox sensor-expressing *T. brucei* cell lines

To obtain parasites that constitutively express the roGFP2 probes in the cytosol or mitochondrion, $4 \times 10^7$ PC cells were transfected with 10 µg of the respective *Not*I-linearized and ethanol-precipitated pHD1991 plasmid by electroporation. After overnight cultivation without selecting antibiotic, 50 µg/ml G418 was added and clones were selected for resistance by serial dilutions as outlined previously (*Musunda et al., 2015*). Expression and correct subcellular localization of the fluorescent redox sensors was verified by live cell imaging.

Cell lines with inducible RNAi against Px I-III, Tpx, or mPrx were obtained by transfecting WT or roGFP2-sensor-expressing cells with *Not*I-linearized pHD678_*pxIII* (*Schlecker et al., 2005*), pHD678_*tpx* (this work), or pHD678_*mprx* (unpublished). Selection was done with 150 µg/ml hygromycin.

## Live cell imaging

$1.2 \times 10^6$ cells were collected and washed with PBS. The roGFP2 fluorescence was examined under a Carl Zeiss Axiovert 200 M microscope equipped with an AxioCam MRm digital camera using the AxioVision software (Zeiss, Jena, Germany).

## Western blot analysis

The sensor-expressing Tpx RNAi cells cultivated in the presence or absence of Tet were harvested and boiled for 5 min in reducing SDS sample buffer. Cell lysates equivalent to $2 \times 10^6$ cells per lane were subjected to SDS-PAGE on 12% gels. Proteins were transferred onto PVDF membranes and developed with antibodies against *T. brucei* Tpx (1:2,000) and mPrx (1:10,000) and secondary HRP-conjugated donkey anti rabbit and goat anti guinea pig antibodies (1:20,000), respectively.

## Plate reader-based in vitro and in vivo measurements of the roGFP2 redox state

The reduced and oxidized forms of roGFP2 have distinct fluorescence excitation maxima (*Dooley et al., 2004*). After excitation at corresponding wavelengths (400 and 485 nm), the emission at 520 nm is measured. The measurements were conducted in 96-well black/clear bottom imaging plates (BD Falcon 353219) in a PHERAstar FS plate reader with build-in injectors (BMG Labtech, Ortenberg, Germany) and optimal focus and gain adjustment. The degree of sensor oxidation (OxD) was calculated using the following formula (*Meyer and Dick, 2010*) and plotted against time.

$$OxD = \frac{I400_{sample} * I485_{red} - I400_{red} * I485_{sample}}{I400_{sample} * I485_{red} - I400_{sample} * I485_{ox} + I400_{ox} * I485_{sample} - I400_{red} * I485_{sample}}$$

I represents the fluorescence emission at 520 nm after excitation at the indicated wavelength, red, fully reduced control and ox, fully oxidized control.

The purified recombinant roGFP2-proteins were diluted in assay buffer (100 mM sodium phosphate, 100 mM NaCl, 50 µM EDTA, pH 7.0) to a final concentration of 1 µM in a total volume of 100 µl per well. To ensure full reduction or oxidation of the control samples, 10 mM DTT or 1 mM diamide was added to the respective well. For oxidation experiments, proteins were pre-treated with 20 mM DTT for 30 min at room temperature and DTT was removed by Zeba Spin Desalting Columns (Thermo Scientific). Except stated otherwise, reduction experiments contained NADPH, TR, and/or hGR in excess to achieve full reduction of the respective thiol. The measurements were done at 25°C.

Parasites constitutively expressing roGFP2, roGFP2-hGrx1, or Tpx-roGFP2 in the cytosol or roGFP2, roGFP2-hGrx1, or roGFP2-Tpx in the mitochondrial matrix were harvested, washed and resuspended in medium without hemin and phenol red. A 200 µl suspension corresponding to $1 \times 10^7$ cells, was transferred into each well and measured at 27°C. For full reduction and oxidation of the sensors, 1 mM DTT and 3 mM diamide, respectively, was added. The respective cells without sensor were used as fluorescence background control. Cells were subjected to exogenous oxidative stresses by injecting different concentrations of $H_2O_2$ or diamide. Untreated cells served as control.

## HPLC analysis of mBBr-labeled thiols

The total cellular free thiols and thiols plus disulfides were determined as described previously for BS *T. brucei* (*Ulrich et al., 2017*). In short, $1 \times 10^8$ PC cells were harvested, washed with PBS, and proteins were precipitated by tri(chloro)acetic acid. The supernatant was split in two samples. The sample for free thiols was directly used for mBBr derivatization. The other one was treated with TCEP followed by labeling with mBBr. The samples were subjected to HPLC analysis and fluorescence detection using a PerfectSil 300 ODS C18 column. Quantification of the thiols was done by comparing integrated peak areas of the sample and respective standards. For calculation of the cellular concentration, values were normalized to the cell number and cell volume, assuming a volume of 96 fl for PC parasites (*Roldán et al., 2011*).

## Biological data collection and data evaluation

All experiments were performed three times on separate days as independent biological replicates, except stated otherwise. The data shown represent the mean ± SD and were evaluated using Graph-Pad Prism (GraphPad Software, La Jolla, CA).

## Acknowledgements

Drs Christine Clayton and Leopold Flohé kindly provided us with the pHD- and pHB 136.1 TbTXNH6-plasmids, respectively. We thank Dr. Koen van Laer for helpful support during the setup of the plate reader experiments. We are grateful to Natalie Dirdjaja for the preparation of recombinant TR and hGR as well as Gsp and T(SH)$_2$. Mattis Hilleke and Verena Schmidtchen are acknowledged for their contributions to the cloning of the mitochondrial expression vectors and the DFMO studies of mitochondrial sensor expressing cells, respectively. This work was supported by a grant from the Deutsche Forschungsgemeinschaft to LKS (Kr 1242/8–1) and by the priority program SPP 1710 of the Deutsche Forschungsgemeinschaft (grant Kr1242/6–two to LKS and grant Di731/3-2 to TPD).

## Additional information

### Funding

| Funder | Grant reference number | Author |
| --- | --- | --- |
| Deutsche Forschungsgemeinschaft | Kr1242/8-1 | R Luise Krauth-Siegel |
| Deutsche Forschungsgemeinschaft | SPP 1710 Kr1242/6-2 | R Luise Krauth-Siegel |
| Deutsche Forschungsgemeinschaft | SPP 1710 Di731/3-2 | Tobias P Dick |

The funders had no role in study design, data collection and interpretation, or the decision to submit the work for publication.

### Author contributions

Samantha Ebersoll, Validation, Investigation, Methodology; Marta Bogacz, Lina M Günter, Investigation, Visualization; Tobias P Dick, Conceptualization, Supervision; R Luise Krauth-Siegel, Conceptualization, Supervision, Funding acquisition

### Author ORCIDs

R Luise Krauth-Siegel https://orcid.org/0000-0003-2164-8116

### Decision letter and Author response

Decision letter https://doi.org/10.7554/eLife.53227.sa1
Author response https://doi.org/10.7554/eLife.53227.sa2

## Additional files

### Supplementary files

- Supplementary file 1. Key resources table.
- Supplementary file 2. The primers used in this work.
- Transparent reporting form

### Data availability

All data generated or analysed during this study are included in the manuscript and supporting files.

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
