## [Decision Letter]

**Acceptance summary:**

The African trypanosomes utilize trypanothione instead of glutathione as their major thiol, and enzymes involved in trypanothione synthesis and metabolism are potential drug targets. Although the functions of trypanothione and associated redox proteins are well characterized, it remains unclear whether trypanothione regulates the redox state of organelles such as the mitochondrion. In this study, the authors have generated a new probe, comprising the redox-sensitive roGFP2 protein coupled to either glutaredoxin or a trypanothione-specific tryparedoxin, to measure the redox state of the cytoplasmic and mitochondrial compartments following treatment of parasites with different oxidants and drugs. Their studies suggest that mitochondria contain a trypanothione-based thiol system and a new oxidoreductase activity capable of sustaining peroxidase activity. The findings provide new insights into the redox biology of these important human pathogens and add to the growing list of applications for which the roGFP sensors are being developed.

**Decision letter after peer review:**

Thank you for submitting your article "A tryparedoxin-coupled biosensor reveals a mitochondrial trypanothione metabolism in trypanosomes" for consideration by *eLife*. Your article has been reviewed by three peer reviewers, including Malcolm J McConville as the Reviewing Editor and Reviewer #1, and the evaluation has been overseen by Dominique Soldati-Favre as the Senior Editor. The following individual involved in review of your submission has agreed to reveal their identity: Ana Tomas (Reviewer #2).

The reviewers have discussed the reviews with one another and the Reviewing Editor has drafted this decision to help you prepare a revised submission.

Summary:

The reviewers found the development of new biosensors for monitoring the redox state of the cytosolic and mitochondrion compartments of the protozoan parasite, Trypanosoma brucei, to be of significant interest. Expression of these biosensors, which included the previously established redox-sensitive roGFP2 protein coupled to either glutaredoxin or a trypanothione-specific tryparedoxin (Tpx-roGFP2), in different cellular compartments, allowed real-time monitoring of the thiol redox state of parasites in response to different oxidants and drugs. The authors show that the probes react preferentially to changes in the trypanothione (T(SH)_2_/TS_2_) redox couple, and present evidence that the trypanosome mitochondrion possess a trypanothione-based thiol system and, an as yet unidentified, oxidoreductase capable of sustaining peroxidase activity. The experimental validation of the sensor is comprehensive and the manuscript well written. The findings provide new insights into the redox biology of these important human pathogens and add to the growing list of applications for which the roGFP sensors are being developed.

Essential revisions:

The reviewers raised a number of points that need to be adequately addressed before the paper can be accepted. Some of the required revisions will likely require further experimentation within the framework of the presented techniques.

1) A question remains around what the new Tpx-roGFP2 probe is measuring (i.e. what drives the reducing and oxidizing half reactions in different organelles of intact cells). For example, while the authors demonstrate that Tpx mediates H_2_O_2_-dependent oxidation processes, they do not assess the extent to which Tpx-roGFP2 becomes oxidized by H_2_O_2_ directly. It would be useful if the authors could (1) test the in vitro reaction of pre-reduced probe with H_2_O_2_ and (2) probe hyperoxidation by 50µM H_2_O_2_ (Figure 3); the diamide control tests reactivity of roGFP2 but not necessarily the TPX moiety which might be hyperoxidized. The probe redox-state could be tested using e.g. the maleimide-based shift assays

2) The authors propose that the *T. brucei* mitochondrion contains an additional oxidoreductase that is dependent on the trypanothione redox couple. However, the data do not exclude the possibility that this oxidoreductase could also or alternatively use a mitochondrial glutathione redox couple. The authors need to address this point.

3) Further to point 1, the impact of the paper would be greatly strengthened if the authors could provide additional information on the identity of the mitochondrial oxidoreductase. The authors note that other studies have identified 21 mitochondrial candidate proteins in *T. brucei* with annotated oxidoreductase activity. Although not essential, confirmation of location or function of these proteins would add to the study.

4) Figure 8C and D. It is noted that "… in cells lacking Tpx, the mitochondrial sensors are still able to respond to exogenously applied H_2_O_2_, in sharp contrast to the situation in the cytosol." Indeed, comparison of these graphs with those of Figure 5C and D, right panels shows a different response. However, the response of the non-induced cells (i.e., maintaining Tpx), is apparently quite different and more marked in the case of the mitochondrial sensors. Could it be that cells expressing the mito probes have inherent higher reactivity and resistance to oxidative stress?

---

## [Author Response]

Essential revisions:The reviewers raised a number of points that need to be adequately addressed before the paper can be accepted. Some of the required revisions will likely require further experimentation within the framework of the presented techniques.

*1) A question remains around what the new Tpx-roGFP2 probe is measuring (i.e. what drives the reducing and oxidizing half reactions in different organelles of intact cells). For example, while the authors demonstrate that Tpx mediates H_2_O_2_-dependent oxidation processes, they do not assess the extent to which Tpx-roGFP2 becomes oxidized by H_2_O_2_ directly. It would be useful if the authors could (1) test the* in vitro *reaction of pre-reduced probe with H_2_O_2_ and (2) probe hyperoxidation by 50µM H_2_O_2_ (Figure 3); the diamide control tests reactivity of roGFP2 but not necessarily the TPX moiety which might be hyperoxidized. The probe redox-state could be tested using e.g. the maleimide-based shift assays*

We incubated the pre-reduced recombinant sensors with 50 µM or 100 µM H_2_O_2_ for 11 min and then added 50 µM or 100 µM TS_2_. Neither Tpx-roGFP2 nor hGrx1-roGFP2 and roGFP2 were oxidized by hydrogen peroxide. All three sensors were oxidized by TS_2_ with the same rate and to the same degree as the respective non-treated probes (Figure 1—figure supplement 2). The data are presented in the new Figure 3—figure supplement 3. This rules out a direct sensor oxidation by H_2_O_2_ as well as a (hyper)oxidation of Tpx. The latter reaction is anyhow not be expected in the case of a protein with a CxxC active site motif.

2) The authors propose that the T. brucei mitochondrion contains an additional oxidoreductase that is dependent on the trypanothione redox couple. However, the data do not exclude the possibility that this oxidoreductase could also or alternatively use a mitochondrial glutathione redox couple. The authors need to address this point.

Our studies do not allow any statement about the presence or absence of glutathione in the mitochondrion. Nevertheless, due to the lack of glutathione reductase, also a mitochondrial glutathione redox system would be directly linked to the trypanothione system. A physiological situation that could give rise to an increase of GSSG, but not TS_2_, is not known. Given a total cellular concentration of ≤ 150 µM, the mitochondrial glutathione concentration is clearly submillimolar. This is in accordance with the finding that the hGrx1-coupled sensor, which is known for its general specificity for the glutathione redox couple, displayed an overall similar and slightly weaker response than the Tpx-coupled sensor when expressed in the cytosol or mitochondrion (Figures 3B and D and 7B and 7D). In trypanosomes, also hGrx1-roGFP2 responds primarily to changes in the T(SH)_2_/TS_2_ ratio. The presence of a mitochondrial glutathione-specific oxidoreductase is highly unlikely.

3) Further to point 1, the impact of the paper would be greatly strengthened if the authors could provide additional information on the identity of the mitochondrial oxidoreductase. The authors note that other studies have identified 21 mitochondrial candidate proteins in T. brucei with annotated oxidoreductase activity. Although not essential, confirmation of location or function of these proteins would add to the study.

A closer inspection of the 21 mitochondrial oxidoreductases described in the proteome analysis of Peikert et al., 2017, yielded only three proteins that might be suitable for the transfer of electrons from the trypanothione redox couple to the mitochondrial tryparedoxin peroxidases. These are Trx1 (Tb927.9.3370), Grx2 (Tb927.1.1770) and a putative glutathione-S-transferase/glutaredoxin (Tb927.7.3500). *T. brucei* Trx1 lacks a mitochondrial pre-sequence, but a genome-wide localization study revealed the protein in the mitochondrion (Dean et al., 2017). Trx1 is inactive in an mPrx assay (unpublished data) but together with human thioredoxin reductase is able to replace the trypanothione/Tpx couple in the Px-assay (Hillebrand et al., 2003). Grx2 also lacks an obvious targeting sequence and has been localized to the mitochondrial intermembrane space (Ebersoll et al., 2018). Grx2 neither replaces nor supports Tpx in the Px-dependent peroxidase assay (Ceylan et al., 2010). Tb927.7.3500 harbors a mitochondrial targeting sequence as well as a CxxC motif but has not been characterized so far. We included a short description of this point in the Discussion.

4) Figure 8C and D. It is noted that "… in cells lacking Tpx, the mitochondrial sensors are still able to respond to exogenously applied H_2_O_2_, in sharp contrast to the situation in the cytosol." Indeed, comparison of these graphs with those of Figure 5C and D, right panels shows a different response. However, the response of the non-induced cells (i.e., maintaining Tpx), is apparently quite different and more marked in the case of the mitochondrial sensors. Could it be that cells expressing the mito probes have inherent higher reactivity and resistance to oxidative stress?

All cell lines expressing the mitochondrial sensors revealed a slightly higher basal OxD, stronger oxidation by hydrogen peroxide and delayed re-reduction when compared to cells harboring the respective cytosolic sensor (compare Figure 7B, D, F with Figure 7B, D, F as well as Figure 5C and D with Figure 8C and D). If the mitochondrial sensors conferred to the cells an inherent higher reactivity and resistance to oxidative stress, one would expect the opposite behavior. Yet, to address this point more explicitly, we treated mito-roGFP2-Tpx-expressing cells and WT parasites with H_2_O_2_. As shown in the new Figure 7—figure supplement 3, both cell types displayed an identical sensitivity.